# Gold catalysts containing interstitial carbon atoms boost hydrogenation activity

Yafei Sun[1,6], Yueqiang Cao [2,6], Lili Wang[1,6], Xiaotong Mu[1], Qingfei Zhao[1], Rui Si[3], Xiaojuan Zhu[1], Shangjun Chen[1], Bingsen Zhang [4], De Chen[5] & Ying Wan [1✉]

Supported gold nanoparticles are emerging catalysts for heterogeneous catalytic reactions, including selective hydrogenation. The traditionally used supports such as silica do not favor the heterolytic dissociation of hydrogen on the surface of gold, thus limiting its hydrogenation activity. Here we use gold catalyst particles partially embedded in the pore walls of meso-porous carbon with carbon atoms occupying interstitial sites in the gold lattice. This catalyst allows improved electron transfer from carbon to gold and, when used for the chemoselective hydrogenation of 3-nitrostyrene, gives a three times higher turn-over frequency (TOF) than that for the well-established $Au/TiO_2$ system. The $d$ electron gain of Au is linearly related to the activation entropy and TOF. The catalyst is stable, and can be recycled ten times with negligible loss of both reaction rate and overall conversion. This strategy paves the way for optimizing noble metal catalysts to give an enhanced hydrogenation catalytic performance.

[1] Key Laboratory of Resource Chemistry of Ministry of Education, Shanghai Key Laboratory of Rare Earth Functional Materials, and Department of Chemistry, Shanghai Normal University, 200234 Shanghai, China. [2] State Key Laboratory of Chemical Engineering, East China University of Science and Technology, 200237 Shanghai, China. [3] Shanghai Synchrotron Radiation Facility, Shanghai Institute of Applied Physics, Chinese Academy of Sciences, 201204 Shanghai, China. [4] Shenyang National Laboratory for Materials Science, Institute of Metal Research, Chinese Academy of Sciences, 110016 Shenyang, China. [5] Department of Chemical Engineering, Norwegian University of Science and Technology, N-7491 Trondheim, Norway. [6] These authors contributed equally: Yafei Sun, Yueqiang Cao, Lili Wang. ✉email: ywan@shnu.edu.cn

Gold is among the least active metals toward molecules at a solid–liquid or solid–gas interface. For example, it has the highest energy barrier for the dissociation of $H_2$ and the least stable chemisorption state[1–3]. This inertness has been attributed to the full filling of the antibonding $d$ state on adsorption and the small orbital overlap with the adsorbate according to the $d$-band model[1,4]. Therefore, when gold is loaded on a less-active support, such as silica and activated carbon, the catalyst shows a very weak catalytic hydrogenation activity due to the fact that the net atomic charge transfer to the substrate or host atoms is very small[5]. Taking into account the fact that activated carbon is the most frequently used carrier for metal catalysts in industry, the activation of, or the charge transfer into gold atoms, which are supported on activated carbon, and its relationship with the catalytic hydrogenation activity are of significant importance but have been seldom reported.

It has been reported that the gold catalysts can sometimes show hydrogenation activity when gold nanoparticles or atoms are supported on reducible oxides[6,7]. In particular, when supported on $TiO_2$, $Fe_2O_3$, etc., gold nanoparticles exhibit high chemoselectivity in reducing a nitro group, whereas other reducing functions are seen in nitroaromatics[8–11]. In these cases, the structure properties of the oxides, including size, phase, exposed crystal plane, oxidation state, reducibility, perimeter sites, etc., have dominant effects on the electronic structure of the nanoparticles[12]. These factors result in the complexity of solid catalysts, and even contradictory results. For example, density functional theory (DFT) calculations show that gold atoms that are neutral or with a net charge close to zero can be active for $H_2$ dissociation, and are usually located at low-coordinated corner or edge positions that do not directly bond to the support. However, the dissociation of molecular hydrogen at the perimeter sites of Au/$TiO_2$[13,14], and a pronounced charge transfer from defect sites on the reduced $TiO_2(110)$ surface to Au clusters have also been reported[6,9,15–17]. The charge transfer calculations, however, are not in good agreement with each other. Some researchers have reported that spectroscopic characterization shows that cationic Au species such as $Au^+$ and $Au^{3+}$ are present at the perimeter interface[18], while negatively charged Au clusters at metal/oxide interfaces have also been confirmed[19]. This complexity limits the correlation of the electronic structure with the catalytic properties, and industrial applications. It has been shown that there is no direct correlation between the number of cationic or anionic Au species and catalytic activity[20].

The motivation for the use of ordered mesoporous carbon as the support is because it is a kind of activated carbon, and porous carbon-supported noble metal catalysts play an important role in the fine chemicals industry, and also have an indispensable role in new energy technologies, such as biomass conversion and fuel cells[21,22]. Their synthesis involves hydrothermal synthesis, polymerization for phenolic resins, and carbonization, all of which have been realized on an industrial scale[23]. The carbonization step with metal species is analogous to the growth of carbon nanotubes by catalytic chemical vapor deposition[24]. Carbon atoms deposit on the transition metal surface such as iron and palladium, and diffuse into the lattice[25]. Schlögl and coworkers reported the formation of interstitial C atoms in a Pd catalyst during the hydrogenation of alkyne, which showed significant promotion in the selectivity for olefins[26]. DFT calculations confirmed the presence of surface and subsurface carbon atoms in Pd[27,28]. Our group found that carburizing a Pd catalyst with interstitial C atoms produces strong inhibition of the adsorption of thiourea, a strong adsorbate[29]. We speculate that the formation of interstitial C in the Au lattice will modify its electronic properties although this has not been reported in experiment[30].

Here, uniform gold nanoparticles with carbon atoms occupying interstitial sites in the lattice (C–Au), supported on ordered mesoporous carbon have been synthesized. Combining X-ray absorption spectroscopy (XAS) with X-ray photoelectron spectroscopy (XPS) analysis, we are able to measure the $d$-electron gain of Au. There is a linear relationship between the $d$-electron gain of gold and its activation entropy ($\Delta S^{0*}$), and catalytic

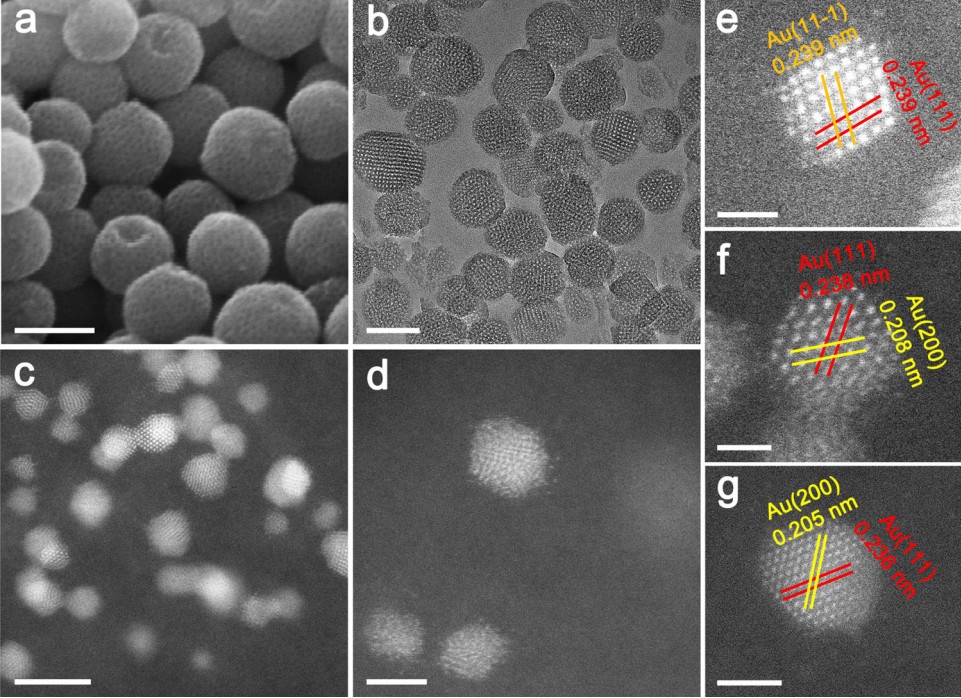

**Fig. 1 Electron micrographs. a** Representative HRSEM image, **b** TEM image of an ultrathin section, **c**, **d** HAADF-AC-STEM images for fresh C–Au-2.4/OMC; AC-STEM images for fresh **e** C–Au-1.6/OMC, **f** C–Au-2.4/OMC, and **g** C–Au-3.9/OMC. The scale bar is 100 nm for (**a**, **b**), 5 nm for (**c**), 2 nm for (**d**, **g**), and 1 nm for (**e**, **f**).

activity in the chemoselective hydrogenation of the nitro group in substituted nitroaromatics. The electron gain of 0.192 of Au for the C–Au-1.6/OMC catalyst is much higher than that of the well-known Au/TiO$_2$ catalyst, showing an almost three times higher turn-over frequency (TOF) value than the latter. This strategy paves the way for optimizing noble metal catalysts to give them enhanced hydrogenation catalytic performance.

## Results

**Structure of supported gold interstitial nanocatalysts.** A series of gold interstitial nanocatalysts supported on ordered mesoporous carbonaceous materials (C–Au/OMC) with different gold nanoparticle sizes was produced by hydrothermal synthesis, polymerization of the phenolic resins, and carbonization. Wide-angle X-ray diffraction (WAXRD) patterns show broad diffraction peaks at ~23°, which is attributed to the amorphous carbonaceous framework, and at ~38.02°, which is attributed to the growth of very small Au nanoparticles or the presence of amorphous Au species[31] (Supplementary Fig. 1). High-resolution scanning electron microscope (HRSEM) images show that all the studied C–Au/OMC catalysts have a spherical morphology with an average diameter of 90 nm (Fig. 1a, Supplementary Fig. 2). Ordered mesopores can be clearly seen on the exposed hemispheres, reflecting an open pore structure on the surface. Nanospheres with open and small mesopores have been verified to eliminate the diffusion limitation especially for reactions involving large molecules[31]. A transmission electron microscope (TEM) image (Fig. 1b) taken from an ultrathin section of the C–Au-2.4/OMC shows spherical carbon particles about 100 nm in diameter that appear to contain ordered mesopores (Supplementary Fig. 3). Gold nanoparticles with a uniform size are located inside the structure, and their size is estimated to range from 1.6 to 9.0 nm. Interestingly, semi-exposed gold nanoparticles can be clearly observed. The similar results have also been found in AuPd nanoparticles partially embedded in the pore walls of ordered mesoporous carbon[32]. Nanoparticles with uniform sizes can also be clearly observed in large areas in high-angle annular dark-field spherical aberration corrected-scanning transmission electron microscopy (HAADF-AC-STEM) images (Fig. 1c, d and Supplementary Fig. 4). HAADF-STEM images were also taken for C–Au-2.4/OMC (Supplementary Fig. 5). There are more bright particles in the centers of the circles than at the edges, implying the presence of nanoparticles inside the spheres. It should be mentioned that isolated gold atoms or dimers are not present even with a particle size as small as 1.6 nm.

Therefore, the catalytic activity is exclusively due to the nanoparticles. The diffractogram of the Au particles, shows an increasing (111) spacing from 2.35 Å in a typical pure Au foil, to 2.36, 2.38, and 2.39 Å in the studied catalysts with Au particle sizes of 3.9, 2.4, and 1.6 nm, respectively, as measured from the AC-STEM images (Fig. 1e–g). It is well-known that lattice contraction occurs in small nanoparticles or clusters, and is related to a higher proportion of outer shell undercoordinated atoms with respect to fully coordinated inner shell atoms and is proportional to the reciprocal of the particle size (2.75% contraction for 2.0 nm nanoparticles)[33]. Therefore, increased lattice fringe separation for the C-modified gold nanoparticles might be related to the dissolution of C atoms in Au, and their occupation of interstitial sites in the Au lattice. This effect increases with smaller particle size.

Although carbon solubility in Au is low, $sp^2$-bonded structures may nucleate on the Au surface for the growth of low-dimensional carbon materials including carbon nanotubes, graphene, etc.[34,35]. In fact, gold has a eutectic alloy phase diagram with carbon, and the carbon concentration in bulk Au at the eutectic point is ~0.08 at%[36]. Specifically, the C adatom diffusion barrier is low for Au, and a significant acceleration of the diffusion step can be achieved when a monoatomic carbon precursor (e.g., CH$_4$) is used[37]. In the present synthesis, the decomposition of phenolic resins and a triblock copolymer could generate small molecules including CH$_4$ at elevated temperatures[32]. The carbon atoms might preferentially precipitate on the Au particle surface because the surface tension of graphite is smaller than that of Au[38]. As a consequence, carbon atoms continuously adsorb on, and quickly diffuse into the surface of the Au catalyst[34]. The carbon solubility in the particle is inversely related to its size, and is estimated to increase by 12 times as the particle size decreases from 9.0 to 1.6 nm[34]. This solubility apparently correlates with the lattice expansion as measured from the AC-STEM images, confirming carbon dissolution in the Au lattice. It should be mentioned that the observed lattice expansion is small possibly due to the relatively low C solubility and the intrinsic lattice contraction in small size particles. However, a continuous exchange of C atoms in Au between surface and subsurface has been reported so that a dynamic equilibrium is established. The mobility of carbon in the lattice may cause residual vacancies and in turn produce a relatively uniform lattice expansion[39]. Indeed, on one occasion an increasing concentration of CH$_4$ was fed into the furnace, and graphene sheets were formed and covered the Au surface (Supplementary Fig. 6).

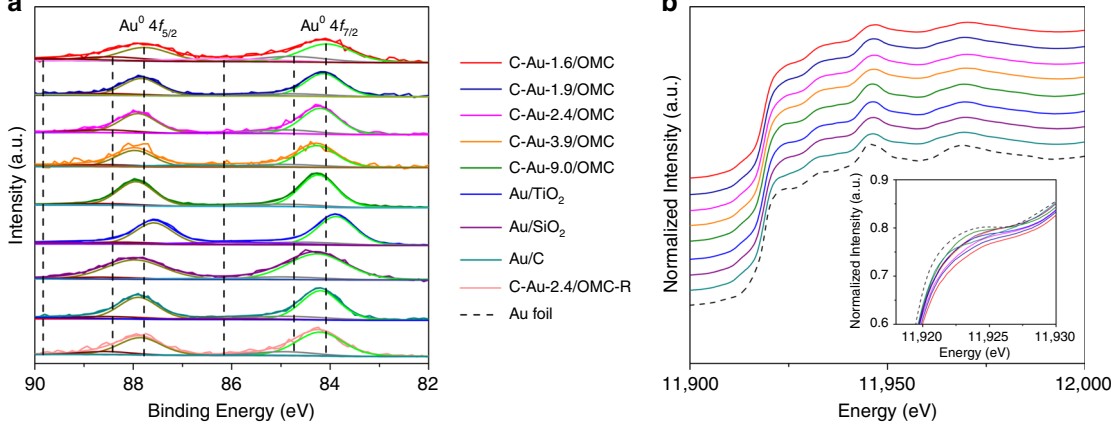

**Fig. 2 Electronic properties. a** XPS spectra of the Au 4$f$ level; and **b** XANES spectra of the Au $L_3$-edge for Au nanocatalysts containing interstitial carbon supported on ordered mesoporous carbon. C–Au-2.4/OMC-R represents the catalyst after being used in five runs. For comparison, commercial Au/TiO$_2$, Au/SiO$_2$, and Au/C were also investigated.

Pseudo-type-I $N_2$ sorption isotherms with capillary condensation at low pressures are detected for all the catalysts studied, similar to the isotherms of pristine mesoporous carbon nanospheres and typical of the isotherms of small mesopores (Supplementary Fig. 7). The pore volumes and the Brunauer–Emmett–Teller (BET) surface areas are close to those for pristine OMC and are ~0.56 $cm^3 g^{-1}$ and 594 $m^2 g^{-1}$ (Supplementary Table 1), respectively.

For comparison, commercial supported catalysts including Au/$TiO_2$, Au/$SiO_2$, and Au/C were also used as catalysts, with gold particle diameter mode values of 4.1, 5.8, and 6.3 nm (Supplementary Fig. 8, and Table 1), respectively. The gold nanoparticles are loaded on the titania surface and inside the pores of porous silica and carbon.

**Electronic properties**. Figure 2a shows measured and fitted Au $4f$ XPS spectra for Au nanocatalysts, and the major peaks for $Au^0$ in all can be fitted with minors of $Au^+$ and $Au^{3+}$, which may originate from undercoordinated sites[39]. There are many possible reasons for the shifts in binding energy. The carrier effect was first investigated by comparing the shifts in the $4f_{7/2}$ level for commercial Au/$TiO_2$, Au/C, and Au/$SiO_2$, which have similar gold particle sizes with bulk Au. The binding energy value for Au/$TiO_2$ is lower than for the bulk metal as has been well documented[40]. When the substrate has localized $p$ or $d$ orbitals with binding energies that overlap those of the cluster $d$ orbitals, a reduction in binding energy appears due to charge transfer from the surface to the Au clusters[17,41]. In contrast, this value is higher for the clusters when the metal-support interaction is weak[41]. It is true that the Au $4f_{7/2}$ binding energies increase for commercial Au/C and Au/$SiO_2$. It should be mentioned that this simple isolation effect is an approximation, and is only used to explain the positive shift for Au supported on a carbonaceous support compared to metallic gold. The size effect is then considerable for the gold nanocatalysts containing interstitial C. The C–Au-9.0/OMC catalyst shows a binding energy for Au $4f_{7/2}$ of 84.24 eV, close to that for commercial Au/C. The value increases to 84.27 eV for C–Au-3.9/OMC. This shift may be attributed to the particle size effect, which is a final state effect according to the size-dependent electrostatic interaction between the cluster and an escaping photoelectron. It has also been reported that there is a negative core-level shift for the surface atoms of macroscopic Au due to an initial state effect, which is $6s \rightarrow 5d$ charge reorganization for the more undercoordinated surface atoms; but for nanoclusters, the initial state shift is mainly overcompensated by the electrostatic final state effect[42]. The present positive shift is in good agreement with the fact that there are shifts to a higher energy for clusters compared to that of the bulk metal. The change is determined by the surface-to-volume ratio, and the photoemission onset is influenced by an initial state effect involving charge transfer[41,43]. However, the binding energy shift does not follow a monotonic increase when further reducing the Au particle size. The Au $4f_{7/2}$ binding energy shifts unexpectedly to 84.20 eV for the 2.4 nm Au, and finally to 84.07 eV for the 1.6 nm Au. This distinctive reversal of the binding energy shift in the Au $4f$ doublet as a function of Au particle size has been observed in Au/$TiO_2$ with similarly small changes, and may be assigned to a combined contribution of the charge transfer from surface to clusters, the initial state effect and the electrostatic final state effect[17]. In the present case, C diffusion that leads to a rearrangement of electron density may be dominant for the charge transfer. Viñes et al. found that the $3s$ core levels of Au around C shifts the binding energy to a higher value compared to a pristine Au(111) surface with no interstitial C, which is an evidence for the redistribution of orbitals[30]. The high energy XPS should be used to give more distinct and direct

evidence on interstitial C in future studies. Here a charge transfer either from the carbon atom or from the $s$, $p$ electron redistribution of Au may be reasonable.

The X-ray absorption near-edge structure (XANES) spectra of all the studied gold catalysts at the $L_3$-edge exhibit three similar peaks at about 40 eV above the edge, indicating a face-centered cubic ($fcc$) structure in the nanoparticles[44] (Fig. 2b, Supplementary Fig. 9). The first resonance at the edge arises from $2p_{3/2} \rightarrow 5d_{5/2}$ and $5d_{3/2}$ dipole transitions in the vicinity of the Fermi level, the intensity of which is related to the unoccupied densities of $d$ states ($d$-hole counts)[45]. Weak resonance can be detected in an Au foil due to $s$-$p$-$d$ hybridization, although the $5d$ orbitals in Au atoms are nominally full. The supported gold nanoparticles show a decrease in the intensity of this resonance. This could be partially due to the nanosize effect[44]. The $s$-$d$ hybridization is increased by a stronger $d$–$d$ interaction in the nanoparticles, leading to an increase of $d$-electron count at the Au sites[44]. When comparing the intensity of the first resonance for C–Au-3.9/OMC, commercial Au/$TiO_2$, Au/$SiO_2$, and Au/C, which have similar nanoparticle sizes, one observes an obvious difference. This result indicates a change in the surface environment in these catalysts in addition to the nanosize effect. Au/$TiO_2$ has been reported to have a lower resonance intensity than Au particles of a similar size in the absence of strong interaction with surface capping molecules[46]. The metal-support interaction results in a negative charge injection into the metal particles by reducible oxides, and a higher increase in $d$-electron density at the gold site compared to bare Au nanoparticles[47]. The number of unoccupied Au $5d_{5/2}$ states near the Fermi level is depleted. The interstitial C in the Au lattice for C–Au-3.9/OMC can thus account for the further decrease in the white line intensity compared to Au/$TiO_2$, and the increase of the $d$-electron density. In addition, the white line intensity for C–Au-9.0/OMC with a particle size of 9.0 nm is even lower than that of commercial Au/C with a particle size mode of 6.3 nm, again indicating that the interstitial C accounts for the electron/hole density at the $5d$ band, and modifies the electronic behavior of the gold nanoparticles. The area under the white line can be used to count the $d$-charge hole redistribution[48]. A semi-quantitative calculation was made for the $d$-electron gain of the catalysts, which follows the order: C–Au-1.6/OMC > C–Au-1.9/OMC > C–Au-2.4/OMC > C–Au-3.9/OMC > Au/$TiO_2$ > C–Au-9.0/OMC > Au/C > Au/$SiO_2$ (Supplementary Fig. 10, Supplementary Table 2).

The net charge transfer is extremely low (<0.01 e) for commercial Au/$SiO_2$ and Au/C with gold particle sizes of 5.8 and 6.3 nm, respectively, and fewer active carriers, in good agreement with the literature[49]. The increased charge transfer for C–Au-3.9/OMC with similar particle sizes is exclusively related to the Au lattice doping by C. In addition, a charge transfer of 0.192 electrons for the 1.6 nm gold nanoparticles with ~120 atoms was measured. The $d$-electron gain is reduced to 0.061 electrons for the C–Au nanocatalyst with 9.0 nm gold nanoparticles. The energy levels and the electron occupancy of the valence-shell atomic orbitals of a surface atom will be influenced by the neighboring atoms in the lattice[26]. The localized $e_g$ band in $d$ orbitals with its high density of states is more readily filled than the metallic $t_{2g}$ band by the introduction of $s$ electrons from the interstitial atoms[50]. As a result, a collective contribution from the particle size effect and the charge transfer from carbon or $s$, $p$ electrons to $d$ electrons of Au in the present C-modified Au nanocatalysts is assigned to the control of the electron/hole density at the $d$ band, and interstitial carbon is more dominant. Lastly, the charge transfer for Au/$TiO_2$ is mostly due to the interface interaction. The electron counting is similar to studies based on the well-defined Pt-group catalysts[32,51–53]. A maximum of 0.11 electrons per Pt atom appears for the Pt deposition with

30–70 atoms on a well-defined CeO₂(111) surface[53]. The peaks belonging to the higher energy part, are structure dependent, and the intensity can be related to the change in the surrounding atoms for each Au atom[54]. The shape for the C–Au/OMC samples is close to that of the Au foil, but the intensity is much lower. The C–Au-1.6/OMC has the minimum intensity. Both the decrease in the number of adjacent Au atoms and the increase in the number of interstitial carbon atoms may contribute to reducing the intensity.

**Selective hydrogenation of 3-nitrostyrene over C–Au/OMC.** The catalytic hydrogenation of aromatic nitro compounds is a cleaner option to produce anilines, but the low selectivity when other reducible groups exist in the reactant molecule is still a drawback, due to the strong adsorption of unsaturated groups on the catalyst surface[55]. Increased activity has been demonstrated on Au/TiO₂ for the conversion of nitrostyrene by Corma et al., but the hydrogenation of nitrostyrene by gold nanoparticles supported on activated carbon or silica shows a very low conversion, and produces both the reduction of the C=C double bond and nitro functional groups[11]. Therefore, the selective hydrogenation of 3-NS to 3-vinylaniline (3-VA) was chosen as the model reaction for the investigation of the catalytic performance of mesoporous carbon-supported gold nanocatalysts.

The kinetic results in Fig. 3a clearly show that with C–Au/OMC catalysts, the conversion plots are almost linear with reaction time for the selective hydrogenation of 3-NS. The TOF value follows the order: C–Au-1.6/OMC > C–Au-1.9/OMC > C–Au-2.4/OMC > C–Au-3.9/OMC > C–Au-9.0/OMC (Supplementary Fig. 11). The reference Au/TiO₂ catalyst shows a similar high activity and selectivity values to those reported in the literature[6,56]. Notably, the TOF value for Au/TiO₂ is lower than for C–Au-3.9/OMC with a similar particle size, while commercial Au/C and Au/SiO₂ catalysts, which have Au particle size modes of 6.3 and 5.8 nm, respectively, show almost no catalytic activity. Therefore, the particle size is possibly not the decisive factor in determining the activity. The selectivities for C–Au-1.6/OMC, C–Au-1.9/OMC, and C–Au-2.4/OMC are high, reaching ~100% at a high conversion (Supplementary Fig. 12). For C–Au-3.9/OMC and C–Au-9.0/OMC, when the conversion is above 80%, the only detectable by-product (<5%) is 3-ethylaniline (3-EA), and others, including 3-ethylnitrobenzene and azoxystyrene, are undetectable, analogous to the results over Au/TiO₂ nanocatalysts reported by Corma et al.[6].

No diffusion phenomena were observed when stirring at a rate of over 500 rpm in the hydrogen reactor (Supplementary Fig. 13). In addition, gold nanocatalysts were synthesized with similar particle sizes (2.4 nm) but different gold loadings (0.5–0.8 wt%, Supplementary Fig. 14). Under the conditions we used we have proved that the initial rate of the reaction increases linearly with the concentration of gold in catalysts with similar gold nanoparticle sizes (Supplementary Fig. 15). As a result, diffusion limitation can be excluded[57,58]. The leaching of gold species from the solid catalyst, which can catalyze the hydrogenation has been argued by some researchers. A solid quenching test was performed using mesoporous silica SH-SBA-15 containing mercapto functional groups as the trapping agent. Once the metal species leach into solution, mercapto functional group will trap them, and quench the catalysis by leaching metal species in liquid reactions[59]. The unchanged hydrogenation activity and selectivity in the presence of SH-SBA-15 demonstrates that there is no leaching of gold with the present C–Au/OMC catalysts regardless of Au particle size, and solubilized gold in solution during the reaction and redeposition on the carrier surface after the reaction can be excluded (Supplementary Fig. 16). The reusability of C–Au-2.4/OMC was also tested. Both the initial reaction rate ($r_0$) and turn-over number in ten successive runs remained almost the same (Supplementary Fig. 17), indicating the number of active centers does not significantly change. The re-used catalysts were characterized. The XPS spectrum shows no distinct changes compared with the fresh catalyst. The Au concentration in the re-used catalyst after ten runs remained at 0.91 wt%, also similar to that determined for the fresh catalyst. In addition, the average Au particle size for the re-used catalyst also remained almost unchanged, as shown by TEM images (Supplementary Fig. 18). In these cases, the leaching and/or aggregation of gold nanoparticles can be excluded. A surface reaction on the Au nanoparticles instead of a reaction in solution catalyzed by leaching Au is the best model for the present reaction. By comparison, the commercial Au/TiO₂ catalyst may undergo aggregation during the reaction. After five runs, the gold in the Au/TiO₂ catalyst appears to aggregate during the reaction, with the nanoparticle diameter increasing from 4.1 to 8.0 nm after five runs, and the TOF decreased to 59 h⁻¹ (Supplementary Fig. 11).

To study the kinetics, several experiments were carried out. First, the initial reaction rate was measured at different 3-NS levels under constant H₂ pressure and temperature (Supplementary Fig. 19a). A constant $r_0$ was observed in the range

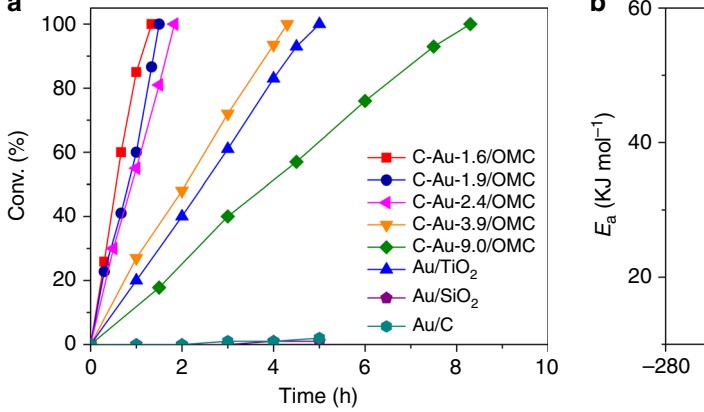
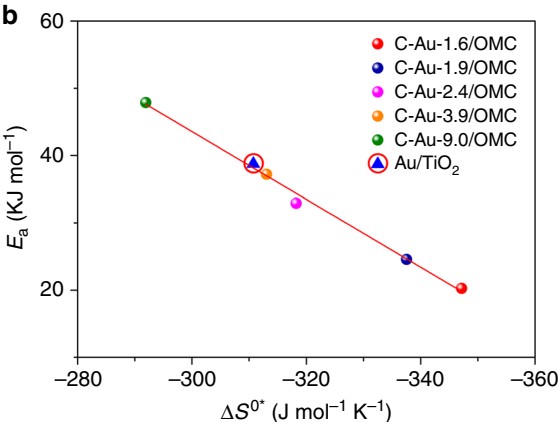

**Fig. 3 Kinetics analysis. a** 3-NS conversion as a function of time with the following reaction conditions over C–Au nanoparticles of different sizes supported on ordered mesoporous carbon. The reaction conditions were: 0.78 μmol of Au; 0.41 mmol of substrate; 5 mL of ethanol; 140 °C; 4.0 MPa H₂. **b** Relationship between the activation energy ($E_a$) and entropy of activation ($\Delta S^{0*}$) for the hydrogenation of 3-NS to 3-VA. For comparison, the results over commercial Au/TiO₂, Au/SiO₂, and Au/C are also given.

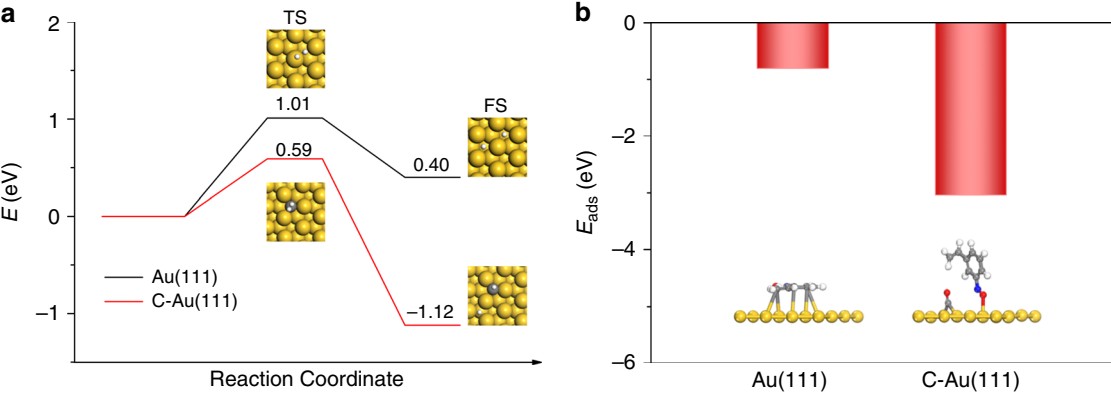

**Fig. 4 DFT calculations. a** Energy profiles for the dissociation of H₂ at Au surfaces. The black line belongs to heterolytic dissociation on clean Au(111), and the red one to heterolytic dissociation on C-modified Au(111). The zero-point energy correction is included in the profile. **b** Adsorption modes of 3-NS and its corresponding energies on clean Au(111) and C-modified Au(111).

$0.0514$–$0.095\ \mu\mathrm{mol\,L^{-1}}$, implying that the adsorption/desorption equilibrium of 3-NS has a minor effect on the kinetics. This result, together with the fact that linear 3-NS conversion vs time plots for all studied catalysts were observed as mentioned above, suggests a zero order for 3-NS in the rate equation. Second, experiments were performed at different initial hydrogen pressures while keeping the 3-NS concentration, the reaction temperature and the amount of catalyst constant. These showed a linear increase in $r_0$ with increasing H₂ pressure within the range of pressures studied (Supplementary Fig. 19b). Third, the rate constant ($k$) for 3-VA formation with H₂ was compared with that using D₂ to investigate the rate-determining step. The experimentally observed $k_H/k_D$ ratio is ~2.7. This obvious isotopic effect confirms H₂ dissociation as the rate-determining step. All these results are in good agreement with those in the literature for Au/TiO₂[60]. 3-NS and H₂ react after adsorbing on active sites with the same nature, but H₂ dissociation is the rate limiting step of the reaction (Supplementary information methods). The overall reaction is first order for H₂, at least for high concentrations of 3-NS. The Arrhenius plots shown in Supplementary Fig. 20 were obtained by calculating $k$ at different reaction temperatures. The apparent activation energies ($E_a$), and the apparent entropies changes $\Delta S^{0*}$ for the catalysts were estimated based on the Arrhenius equation and transition state theory (Supplementary information methods). The activation entropy reflects the freedom of the system, using the hypothesis that catalysis is related to the surface intermediate with the promotion of an electron from the adsorbate to a localized band of the solid[61]. The increase in the binding energy of the molecule or functional group to the surface leads to a greater restriction on vibrational and rotational freedom[62]. A linear relationship is observed for $E_a$ and $\Delta S^{0*}$ for the C–Au/OMC catalysts with various nanoparticle sizes, which is called the compensation effect (Fig. 3b)[63]. This result implies a relationship between the adsorption configuration and electronic properties, and the catalytic activity. Notably, the commercial Au/TiO₂ catalyst follows the same trend.

**DFT calculations**. The diffusion of C atoms into interstitial sites of the Au lattice was also examined by DFT calculations. The most favorable absorption configuration and energies for a single carbon atom on the close-packed Au(111) and stepped (211) facets are summarized in Supplementary Figs. 21–22. Carbon atoms preferentially adsorb on *fcc* threefold hollow sites on the surface and tetrahedral interstices on the subsurface for Au(111). Hexagonal close-packed(*hcp*) threefold hollow sites and tetrahedral interstices are the most preferred by C species on the Au

(211) surface and subsurface, respectively. Since Au is the most noble metal and has an extremely high dissociation energy barrier for H₂, we first compare the dissociation of H₂ on a clean Au(111) surface with one modified with C (Fig. 4a). As expected, on clean Au(111), one H atom is above the Au atom, and the other H atom is at the site bridging two Au atoms. The energy barrier of H₂ dissociation of 1.01 eV and the adsorption energy, with respect to H₂, is around 0.40 eV, close to the reference value[64]. When carbon diffuses into the Au(111) surface, the H₂ can be activated only by the C atom, and both H atoms are adsorbed on the C atom, reducing the activation barrier to 0.59 eV, implying a lower energy process. The final state shows one H atom being anchored on a carbon atom, and the other on an Au atom, with the adsorption energy being little changed. These results indicate that the heterolytic dissociation of H₂ by Au and C is dominant for the hydrogenation, similar to that by perimeter sites over Au/TiO₂[14]. It should be mentioned that the subsurface carbon atom on Au(111) diffuses to the surface as a result of H adsorption, in good agreement with the dynamic mobility of C atoms on transition metal (111) surfaces under working conditions of low C coverage[39]. As a result, the presence of subsurface C would facilitate the mobility of carbon in the lattice, possibly to some C-free areas of the surface[39]. This would further enhance the H adsorption.

DFT calculations show that parallel adsorption with the shortest average distance between adsorbate the Au(111) and Au(211) surfaces is most favorable (Fig. 4b, Supplementary Fig. 23). The chemical bonding between the functional groups of nitro, phenyl, and vinyl group in 3-NS and clean Au(111) and Au(211) surfaces is negligible. The calculated adsorption energies are low, primarily due to physical adsorption, in good agreement with nitrobenzene adsorption on Au surfaces[15]. This result predicts that 3-nitrostyrene interacts weakly with commercial Au/C and Au/SiO₂ in which gold nanoparticles have weak interactions with the less-active supports, and the selectivity is low for the hydrogenation of nitro groups and the C = C double bond due to negligible preferential adsorption toward one or the other, in agreement with experiment. A completely different situation occurs when 3-NS interacts with Au surfaces containing interstitial C. The most favorable adsorption configuration for 3-NS over the C–Au(111) and C–Au(211) surfaces is perpendicular adsorption. On the C–Au(111) surface, one of the N–O bonds breaks with O on the carbon atom, and the other O on the nonneighboring Au atom, while the dissociation adsorption of N–O bonds on the C–Au(211) surface occurs on a carbon atom and the adjacent Au site. The conformation of the N atom shows a noticeable distortion from planarity. Selective adsorption and dissociation of the nitro group appear to occur naturally.

## Discussion

It is obvious that interstitial C can be produced in the Au lattice during the carbonization of an ordered mesoporous carbon carrier, and plays a vital role in selective hydrogenation. First, $H_2$ molecules can be activated. The heterolytic dissociation of $H_2$ can take place on C–Au due to electron transfer between C and Au. The adsorption configuration also implies a significant effect of the electronic properties of C–Au on the chemisorption. This selective adsorption and activation on Au surfaces containing interstitial C compared to pure surfaces explains the high chemoselectivity observed for the C–Au/OMC catalysts compared to commercial Au/C and Au/SiO2. In addition, a continuous exchange of C atoms between surface and subsurface possibly occurs, which may facilitate the adsorption of $H_2$[39]. The dynamics of C movement in the Au lattice may also be responsible for the desorption of products to inhibit accumulation on the surface, which enhances the reusability. The changes in solid state properties merely reflect the existence of a bound state for the absorbed intermediate that is energetically most favored for the catalytic reaction, and the entropy change is highly associated with this process. The inherent electronic energy for the entropy changes must involve the extent of occupation of the various bands[65]. As a result, the activation entropy and activity changes can be related to the distribution of $d$-band holes between the $t_{2g}$ and $e_g$ bands. A linear relationship has been plotted for the calculated changes in the activation entropy and TOF values vs the electron gain for the Au $d$ orbital (Fig. 5). The activation of the noble Au surface is achieved by the injection of $d$ electrons. It should be mentioned that C–Au-3.9/OMC has a larger $d$-charge gain than Au/TiO2 with a similar particle size. When comparing the TOF values for C–Au-3.9/OMC and Au/TiO2, they follow the same trend. The catalyst with the highest $d$-electron gain has the highest TOF ($670\,h^{-1}$), almost three times that of a commercial Au/TiO2 catalyst and a >99% selectivity to 3-VA. By comparison, commercial Au/C and Au/SiO2 catalysts with an extremely low $d$-electron gain are almost inert.

In summary, an attempt has been made to boost catalytic hydrogenation by using gold nanocatalysts with C occupying interstitial sites in its lattice, which contributes to the electron transfer or redistribution at Au sites. These C–Au catalysts with $d$-electron gain show a high chemoselective hydrogenation activity of the nitro group in 3-nitrostyrene even when supported on a less-active ordered mesoporous carbon. This strategy paves the way for the design of noble metal catalysts loaded on an industrially available carbon carrier with unique chemoselectivity in organic reactions.

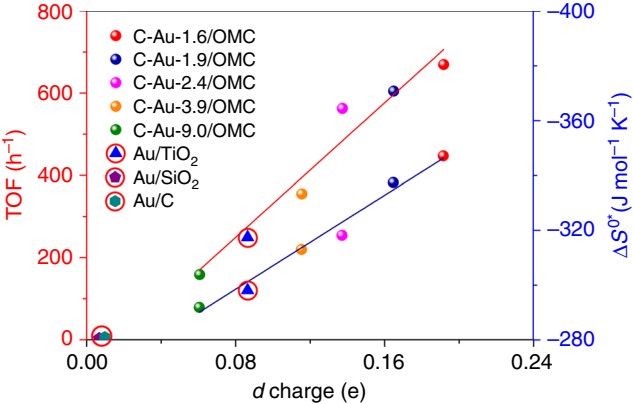

**Fig. 5 Descriptor.** The relationship between the $d$-electron gain at an Au site and the turn-over frequency (TOF, red line) and the entropy of activation ($\Delta S^{0*}$, navy line) for the selective hydrogenation of 3-NS.

## Methods

**Synthesis of C–Au/OMC**. Au catalysts encapsulated in ordered mesoporous carbon nanospheres were synthesized using a hydrothermal method. In a typical synthesis, 0.6 g phenol (Macklin, AR), 15 mL NaOH (0.1 M), and 2.1 mL formalin aqueous solution (37 wt%) were mixed at 70 °C under a stirring speed of 360 rpm for 30 min. Then, a clear aqueous solution of 0.96 g F127 (Acros Chemical Inc., $EO_{106}PO_{70}EO_{106}$, $M_W = 12,600\,g\,mol^{-1}$) in 15 g Milli-Q water was added to the solution, and the mixed solution was stirred at 67 °C for further 2–4 h. After that, a solution prepared by mixing 50 g Milli-Q water, 1.5 mL HAuCl4 (gold concentration in water: 24.3 mmol $L^{-1}$) and 0.128 g 3-mercaptopropyltrimethoxysilane (MPTMS, 85 wt% Acros Chemical Inc.) were added. The above reaction was immediately stopped by cooling the solution to room temperature when precipitation was observed in the solution. When the precipitate was dissolved after a short time of quiescence, the obtained solution was transferred into several autoclaves, diluted with $H_2O$ (3.16 mL of water per mL of solution) and heated at 130 °C for 24 h. A yellow solid was collected after centrifugation, washing with Milli-Q water and ethanol, and dried in a vacuum overnight at 80 °C. After carbonization at 700 °C for 3 h under high-purity $N_2$, which removed the triblock copolymer templates and carbonized the resins, and cooling at 200 °C for 6 h in air a black product was obtained, which is denoted C–Au-$n$/OMC, wherein $n$ represents the size of the gold particles measured by TEM. By changing the ratio between HAuCl4 and MPTMS and the carbonization temperature while keeping the other processing parameters constant, catalysts with gold particles ranging from 1.6 to 9.0 nm diameter were obtained (Supplementary Table 3). Customized reference Au catalysts including Au/TiO2, Au/SiO2, and Au/C were purchased from Haruta Gold Inc. (Japan).

**Characterization of the materials**. The crystal structure of the catalysts was analyzed by WAXRD, which was performed on a Rigaku D max-3C diffractometer using Cu Kα radiation. The morphology of the catalysts was characterized by HRSEM using a Hitachi S-4800 ultrahigh resolution cold FEG with an in-lens electron optic operating at 20 kV, and by TEM using a JEM 2100 microscope operating at 200 kV. Lattice resolution images of the Au catalysts were taken by HAADF-STEM, using a JEM-ARM 200 F microscope. The AC-STEM images were collected using a Titan Cubed Themis G2 300 microscope operated at 200 kV. A Micromeritics TriStar II 3020 analyzer was used to measure the $N_2$ adsorption–desorption isotherms of the catalysts at 77 K. The specific surface areas ($S_{BET}$) and the pore size ($D_P$) distributions of the catalysts were calculated using the BET method and the Barrett–Joyner–Halenda model. A double-crystal Si(111) monochromator was used for energy selection at both the Au $L_3$-edge (11,919 eV). A Lytle detector was used to collect data in the fluorescent mode at room temperature, and XAS data were processed using the Athena program. Individual scans were calibrated and aligned using an Au foil spectrum. Au $L_3$-edge extended X-ray absorption fine structure processing data were fitted using the software package for the inverse fast Fourier transform (IFEFFIT) in R space. XPS was used to characterize the chemical composition and elemental state of the catalysts on a Perkin-Elmer PHI 5000 CESCA instrument. For the XPS studies, a Al Kα source was used. A pass energy of 40 eV and a step size of 100 meV were used for a survey scan. An experimental resolution of 0.5 eV has been fitted from the Ag $3d_{5/2}$ bulk peak. For a detailed analysis, the core-level lines obtained by XPS were numerically fitted by a convolution of a Gaussian and a Lorentzian profile with an additional parameter allowing asymmetry of the line, and the data were calibrated using the C 1s binding energy of 284.6 eV.

The Au content of the catalysts was determined using a Varian VISTAMPX inductively coupled plasma-atomic emission spectrometer.

*Catalytic activity tests*: the hydrogenation reactions of 3-nitrostyrene (Acros, 97 wt%) were performed in a 50 mL autoclave equipped with a Teflon tube (Parr). Normally, the autoclave was loaded with 0.41 mmol 3-NS, 5 mL ethanol, and the C–Au/OMC catalyst and then flushed three times with 0.5 MPa $H_2$ before it was pressurized to the desired $H_2$ pressure of 4.0 MPa and then placed in an oil bath maintained at the reaction temperature of 140 °C. The stirring rate was set to 800 rpm to eliminate any external diffusion. The reaction was stopped at a selected time by immediately cooling the autoclave in an ice-water bath. The solid catalysts were collected by filtration and then washed with Milli-Q water and dried at 80 °C. The collected filtrate was extracted with 20 mL of ethanol. Analysis of the reaction products was done on an Agilent 7890B gas chromatograph equipped with a DB-1 capillary column (30.0 m × 320 mm × 0.25 mm) and a flame ionization detector. Each reaction test was repeated at least three times with an experimental error of ± 5%. Based on the conversion of 3-NS, the yields of 3-VA and 3-EA were given as the results of the reaction. The reaction with deuterium gas was the same as above with the hydrogen replaced by $D_2$. The catalytic activity results were given in terms of the conversion of 3-NS, yield of 3-VA and 3-EA, initial reaction rate (molecules of 3-NS converted per mole of Au per hour), and TOF (molecules of 3-NS converted per surface atom of Au per hour). The TOF value was calculated at a conversion below 25% and was reproducible to within ± 5%. An established method was used to calculate the exposed surface area of the dispersed Au particles[66].

The TOF, $E_a$, $\Delta S^{0*}$, and the elemental reaction steps for the hydrogenation reactions of 3-nitrostyrene over C–Au/OMC catalysts were obtained and are described in the Supplementary information.

*Mass transfer limitation tests*: the tests were carried out under the same reaction conditions as already described. External mass transfer limitations are avoided by using different stirring speeds in the range 600–1000 rpm (Supplementary Fig. 13). The Madon–Boudart tests using 20 mg of C–Au-2.4/OMC catalyst with similar Au dispersions but different Au concentrations ranging from 0.5–1.0 wt% were performed to investigate the internal diffusion limitation. A linear relationship between the initial reaction rate and Au concentration indicated the absence of a mass transfer effect under the reaction conditions (Supplementary Fig. 15).

*Recycling tests*: the recycling hydrogenation reaction of 3-NS was carried out over C–Au-2.4/OMC catalyst under the reaction conditions described above. After each run, the C–Au-2.4/OMC catalyst was recovered after thoroughly washing with copious amounts of ethanol and water, and drying under vacuum at 80 °C overnight. In order to maintain the same amount of catalyst in each run, several parallel reactions were carried out at the same time. The gold contents of the solid as well as in the aqueous solution after each cycle were determined.

*Trapping tests*: thiol group-modified mesoporous silica (SH-SBA-15, Supplementary information) with a molar ratio of SH:Au = 35:1 was used to capture any soluble gold species that had leached into the solution during the reactions and to investigate the stability of the gold catalysts[59].

*DFT calculations*: DFT calculations were carried out with the Vienna ab initio simulation package[67]. The Au(111) and (211) surfaces were modeled using a five-layer metal slab describing a $p(2 \times 2)$ unit cell and a vacuum layer of 12 Å. The relative positions of the atoms are as in the bulk, with an optimized lattice parameter of 4.17 Å, which is close to the previously experimental and the calculated ones (i.e., 4.08 and 4.17 Å, respectively)[68]. Three different surface sites (top, bridge, and hollow) and two different subsurface sites (tetrahedral and octahedral) were considered. More details are presented in the Supplementary information (computational details and Tables 4–8).

## Data availability
The data that support the findings of this study are available from the corresponding authors upon reasonable request.

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

## Acknowledgements

This work was supported by the National Natural Science Foundation of China (21773156, 21503136), the Ministry of Education of China (PCSIRT_IRT_16R49), the International Joint Laboratory on Resource Chemistry of China (IJLRC), the Shanghai Sci. & Tech. and Edu. Committee (17JC1404200) and the Shanghai Gaofeng & Gaoyuan Project for University Academic Program Development.

## Author contributions

Y.W. designed the research, supervised experiments, and edited the paper. Y.F.S., Y.Q.C., and L.L.W. planned synthesis and tested catalysts, analyzed the XAS and kinetics data, and wrote the paper. X.T.M. analyzed the XPS data, performed the XRD and BET characterizations and edited the figures. Q.F.Z. performed the TEM and EDX experiments and data analysis. R.S. performed the XAS experiments and data analysis. X.J.Z. and S.J.C. edited the figures. B.S.Z performed the HAADF-STEM. Y.Q.C. and D.C. contributed to the DFT calculations and discussion on the kinetics. All authors discussed the results and commented on the paper.

## Competing interests

The authors declare no competing interests.
