## [Peer Review File · Nature Communications]

REVIEWER COMMENTS

Reviewer #1 (Remarks to the Author):

Sun et al. show here a really exciting study of synthesis, characterization, use, and reuse of C-promoted Au nanoparticle catalysts for the hydrogenation of 3-nitrostyrene, adjoining experiments with computational simulations, showing the outstanding performance of Au nanoparticles, in activity, selectivity, and stability over time, connected to C direct (facilitating H₂ dissociation and 3-nitrostyrene adsorption) and indirect (filling the Au d-states). The system is a text-book example, but highlights the spectacular importance of considering other species, here interstitial C, on Au, and such details can be a change of paradigm in future research on heterogeneous catalysis. The study seems to be very well carried out, well written, and very complete in many parts. I would gladly recommend publication in Nature Communications, but there are a few aspects, some minor, other major, and one critical, that authors must address before final acceptance.

1. Minor: Authors start stating that Au is the most unreactive, pointing to H₂ chemisorption. By this authors probably wanted to state that is among the least active (not reactive, as Au does not 'react' but catalyzes), plus, in certain conditions, Ag is even less active. See the adsorption energies of H on Au vs. Ag(111) in doi: 10.1016/j.susc.2011.12.017.
2. Major: Authors point on C inside Pd with novel references (24 and 25); but due credit has to be put on much earlier studies stating the subsurface preference (see e.g. <https://doi.org/10.1039/B311054K>, but many others afterwards including coverage and low-coordination effects).
3. Minor: When discussing Fig. 1b, authors state that Au is Im-3m and body-centred cubic, but it is known to be Fm-3m and face-centred cubic (as authors state later). I am sure that it was a mere confusion, and no 'unexplicable' phase transition occurs in Au.
4. Minor: The lattice contraction in metal nanoparticles is well known, but rather than surface/volume, is more related to a higher proportion of outer shell undercoordinated atoms with respect fully coordinated inner shell atoms. That should be clarified.
5. Major: The binding energy (E_b) is claimed to go linear with the coordination number, but should go rather on the 'surface/volume' ratio, express as the lowered-dimensionality reduced particle radius, $r^{-1/3}$, as many other properties do? See <https://doi.org/10.1039/C3CS60421G>.
6. Major: The discussion on the E_b shifts is based on little changes. What is the resolution of the XPS apparatus? Does it permit unequivocally claiming such shifts? Aside, authors should label the two peaks of Au 4f, and explain what would be the low-intensity grey signal in between such when decomposed. Have authors analysed the C 1s? A recent study on 3s levels on Au suggest that having interstitial C shifts the BE to higher values, see <https://doi.org/10.1002/anie.201813037>, which seems to go along the present XPS data.

7. Major: The shifts on the Eb can have multiple origins. Considering size effect, the smaller, the more atom-like are the NPs, and less diffuse electrons, which would make the Eb to increase. However, the charge transfer to (from) Au could decrease (increase) it. However, when having larger surface/volume ratio, one has more undercoordinated sites, and such are known to reduce the BE, as they accumulate there charge density (see doi: 10.1007/s11426-010-0086-z). The discussion should be reconsidered, as, being many factors, one cannot isolate them and rule most out so to explain solely based on one.

8. Minor: Authors suggest a charge transfer from C to Au. This goes along, e.g. with the charge transfer from graphene to Au, but however other reports indirectly suggest positively charged C, although with no Bader charges computed [<https://doi.org/10.1002/anie.201813037>]. That should be discussed.

9. Critical: The authors neglect the ZPE correction, but such should be easily added, as authors should vibrationally characterize the minima and transition state found. This can be also very critical in the H2 dissociation, given its large stretching frequency.

10. Major: The dynamics of the C movement to the surface or its insertion back to the Au matrix could explain the reusability? In this sense, there is a recent study on such which could address the discussion; see doi : 10.1016/j.apsusc.2020.145765.

11. Critical: Perhaps the major critical point is the lack of description of the computational part; it is just succinctly treated, as if it was a mere and basic technique; but the details are necessary for a correct assessment; otherwise, the results could be simply wrong, either because of the model or the method. The list of open questions are, convergence of a property like the adsorption energy with respect the number of layers, basis set kinetic energy and convergence with respect it? K-points density and type, and convergence with respect it? There are more than one single type of hollows (or bridges or tops in the (211) case); have them been all explored? How have the authors treated the adsorption of 3-nitrostyrene; specifically sites and configurations. How to calculate vibrations. The optimization electronic and atomic convergence criteria. The simulation of dispersive forces? What functional have they used and why? How is the smearing treated and details. Types of pseudopotentials. How are isolated atoms or molecules calculated? How are transition state located? With which method and details. All this could well be introduced in a section in the supplementary information.

12. Minor: Put commas between all items in a sequence. In page 12, the sentence with “activated by the only the C atom, and both” reads strange; please amend. In the computational details, it should read p(2x2), lack of dimensionality (better explicit vacuum in Angstroms).

Reviewer #2 (Remarks to the Author):

This is a potentially interesting paper but the catalysis needs a lot of attention. The claim is that by

making the catalyst this way they make a material that has interstitial C in Au nanoparticles supported on carbon. They then carry out a standard model hydrogenation reaction (so there is no novelty in the catalysed reaction). The catalysis they observe with their catalyst is certainly interesting. But the main claim is that their catalyst is far better than gold supported on carbon, silica or titania. But the comparison is with three commercial catalysts and we have no idea how they are made and what they comprise. For a paper at this level this is unacceptable. They need to prepare their own Au comparison catalysts so that the particle sizes are similar and then do a comparison.

The loss of gold can be a real problem and they need to show the data for Au retained on the catalyst after use. Typically quite a lot can be solubilised but it plates out on the reactor and so is not observed in solution.

Finally the English needs attention.

I am not commenting on the microscopy or the DFT calculations

Reviewer #3 (Remarks to the Author):

The authors claim to have supposedly made a catalyst material comprised of gold particles embedded in mesoporous carbon with carbon atoms occupying interstitial sites in the gold lattice. Furthermore, they claim this interstitial doped catalyst is superior to Au/TiO₂ for the chemoselective hydrogenation of 3-nitrostyrene.

Several things concern me about the microstructural characterization aspects of this paper.

Firstly, the Au Particle size distributions are not 'monodisperse' as repeatedly claimed in the text as proven by their own particle size distributions.

Secondly, it has not been proven that the Au particles are actually within the pores of the activated carbon spheres. TEM/STEM is a 2D projection technique unless applied in tomography mode. Indeed, several of the supplementary figures show gold particles in profile on the C surface. Furthermore, if the gold particles were really embedded in the interior of the activated C spheres how do the generated photoelectrons escape in the XPS experiments?

Finally, and even more concerning to me, is the lack of any convincing evidence that C exists as an interstitial species in the f.c.c. Au lattice. Measurement of Au (111) spacings by STEM HAADF imaging over six or so lattice planes to a 0.01Å accuracy is simply not a realistic or feasible proposition. The authors should carefully evaluate the intrinsic errors associated with making such measurements.

Furthermore, even a (very high) 2at% C doping level in a 2nm Au particle would still only mean the incorporation 1-2 interstitial C atoms at most. How would this cause such a uniform expansion over (111) planes in a such 2nm Au nanoparticle?

Responses to Reviewers' comments

Reviewer #1

Comment. Sun et al. show here a really exciting study of synthesis, characterization, use, and reuse of C-promoted Au nanoparticle catalysts for the hydrogenation of 3-nitrostyrene, adjoining experiments with computational simulations, showing the outstanding performance of Au nanoparticles, in activity, selectivity, and stability over time, connected to C direct (facilitating H₂ dissociation and 3-nitrostyrene adsorption) and indirect (filling the Au *d*-states). The system is a text-book example, but highlights the spectacular importance of considering other species, here interstitial C, on Au, and such details can be a change of paradigm in future research on heterogeneous catalysis. The study seems to be very well carried out, well written, and very complete in many parts. I would gladly recommend publication in Nature Communications, but there are a few aspects, some minor, other major, and one critical, that authors must address before final acceptance.

Response. We thank you very much for the positive comments and your reviewing the importance of interstitial C on Au.

Comment 1. Minor: Authors start stating that Au is the most unreactive, pointing to H₂ chemisorption. By this authors probably wanted to state that is among the least active (not reactive, as Au does not 'react' but catalyzes), plus, in certain conditions, Ag is even less active. See the adsorption energies of H on Au vs. Ag(111) in doi: 10.1016/j.susc.2011.12.017.

Response 1. We appreciate this comment. Au is among the least active metals. Accordingly, we have revised our manuscript.

Revised manuscript, Page 3:

“Gold is the most unreactive metal toward molecules at a solid-liquid or solid-gas interface. For example, it has the highest energy barrier for the dissociation of H₂ and the least stable chemisorption state¹.” has been changed to “Gold is among the least active metals toward molecules at a solid-liquid or solid-gas interface. For example, it has the highest energy barrier for the dissociation of H₂ and the least stable chemisorption state^{1,2}.”.

References

2. Ferrin, P., Kandoi, S., Nilekar, A. U., Mavrikakis, M. *Surf. Sci.* **606**, 679 (2012).

Comment 2. Major: Authors point on C inside Pd with novel references (24 and 25); but due credit has to be put on much earlier studies stating the subsurface preference (see e.g. <https://doi.org/10.1039/B311054K>, but many others afterwards including coverage and low-coordination effects).

Response 2. We thank you for this comment. The earlier studies of subsurface C in Pd have been cited in the revised manuscript.

Revised manuscript, Page 4, an earlier reference has been added:

“DFT calculations confirmed the presence of surface and subsurface carbon atoms in Pd^{25,26}.”.

References

26. Yudanov, I. V., Neyman, K. M., Rösch, N. *Phys. Chem. Chem. Phys.* **6**, 116 (2004).

Comment 3. Minor: When discussing Fig. 1b, authors state that Au is $Im\bar{3}m$ and body-centred cubic, but it is known to be $Fm\bar{3}m$ and face-centered cubic (as authors state later). I am sure that it was a mere confusion, and no ‘unexplicable’ phase transition occurs in Au.

Response 3. We are very sorry for this misleading information. Fig. 1b, focusses on the structure of the ordered mesoporous carbon support which we incorrectly said has the body-centered cubic $Im\bar{3}m$ space group. we have removed this discussion.

Revised manuscript, Page 5:

“A TEM image taken from ultrathin sections of the C-Au-2.4/OMC shows a Au nanoparticle size mode of 2.4 nm and well-ordered domains viewed along the [100], [110] and [111] directions, indicating a highly ordered body-centered cubic $Im\bar{3}m$ mesostructure (Fig. 1b).” has been changed to “A transmission electron microscope (TEM) image (Fig. 1b) taken from an ultrathin section of the C-Au-2.4/OMC shows spherical carbon particles about 100 nm in diameter that appear to contain ordered mesopores (Supplementary Fig. 3).”.

Comment 4. Minor: The lattice contraction in metal nanoparticles is well known, but rather than surface/volume, is more related to a higher proportion of outer shell undercoordinated atoms with respect to fully coordinated inner shell atoms. That should be clarified.

Response 4. We agree with the reviewer after re-checking references. The lattice contraction is highly related to a high proportion of outer shell undercoordinated atoms with respect to fully coordinated inner shell atoms. We have accordingly revised our manuscript.

Revised manuscript, Page 5:

“It is well-known that lattice contraction occurs in small nanoparticles or clusters, and is induced by the large surface/volume ratio,” has been changed to “It is well-known that lattice contraction occurs in small nanoparticles or clusters, and is related to a higher proportion of outer shell undercoordinated atoms with respect to fully coordinated inner shell atoms,”.

Comment 5. Major: The binding energy (E_B) is claimed to go linear with the coordination number, but should go rather on the ‘surface/volume’ ratio, express as the lowered-dimensionality reduced particle radius, $r^{-1/3}$, as many other properties do? See <https://doi.org/10.1039/C3CS60421G>.

Response 5. We agree with the reviewer. The shift of the binding energy is attributed to the particle size effect, in good agreement with the fact that E_B shifts to a higher energy for clusters compared to that of the bulk metal, and the change is much related to the surface-to-volume ratio. Accordingly, we have revised our manuscript.

Revised manuscript, Page 8, several references have been added:

“This shift is attributed to the particle size effect, in good agreement with the fact that E_B shifts to a higher energy for clusters compared to that of the bulk metal and the change is a linear function of the average coordination number.” has been changed to “This shift may be attributed to the particle size effect which is a final state effect according to the size-dependent electrostatic interaction between the cluster and an escaping photoelectron. It has also been reported that there is a negative core level shift for the surface atoms of macroscopic Au due to an initial state effect, which is $6s \rightarrow 5d$ charge reorganization for the more undercoordinated surface atoms; but for nanoclusters, the initial state shift is mainly overcompensated by the electrostatic final state effect⁴¹. The present positive shift is in good agreement with the fact that there are shifts to a higher energy for clusters compared to that of the bulk metal. The change is determined by the surface-to-volume ratio, and the photoemission onset is influenced by an initial state effect involving charge transfer^{40,42}.”.

References

41. Zeng, Z., Ma, X., Ding, W., Li., W. *Sci. China Chem.* **53**, 402 (2010).

42. Viñes, F., Gomes, J. R. B., Illas, F. *Chem. Soc. Rev.* **43**, 4922 (2014).

Comment 6. Major: The discussion on the E_B shifts is based on little changes. What is the resolution of the XPS apparatus? Does it permit unequivocally claiming such shifts? Aside, authors should label the two peaks of Au 4*f*, and explain what would be the low-intensity grey signal in between such when decomposed. Have authors analysed the C 1*s*? A recent study on 3*s* levels on Au suggest that having interstitial C shifts the BE to higher values, see <https://doi.org/10.1002/anie.201813037>, which seems to go along the present XPS data.

Response 6.

- (1) We agree that the E_B shifts are small. For the XPS studies, a Al K_α source was used. A pass energy of 40 eV and a step size of 100 meV were used for a survey scan. An experimental resolution of 0.5 eV has been fitted from the Ag $3d_{5/2}$ bulk peak. For a detailed analysis, the core-level lines obtained by XPS were numerically fitted by a convolution of a Gaussian and a Lorentzian profile with an additional parameter allowing asymmetry of the line, and the data was calibrated using the C 1*s* binding energy of 284.6 eV.
- (2) Such a small change has also been observed by Huang and co-workers¹. The Au $4f_{7/2}$ binding energy shifts from 84.13 eV for 1.0 Å-thick Au film on a reduced TiO₂(110) surface, to 84.06 eV for 1.5 Å Au and finally to 84.13 eV for 2.0 Å Au. The authors pointed out that this was the first experimental evidence for the charge transfer from the defective sites on the reduced TiO₂(110) surface to Au clusters. However, as mentioned by the reviewer, the unequivocal conclusion on such shifts may be exaggerated. In order to not mislead the readers, we have revised our manuscript to avoid this.
- (3) The low-intensity grey signals in the XPS spectra of the Au 4*f* level belong to Au⁺ and Au³⁺ which may originate from undercoordinated sites (Fig. 1a).
- (4) We have analyzed the C 1*s* spectra as shown in Fig. 1b. But it should be mentioned that the carrier is a kind of activated carbon, which will have an effect on the C 1*s* spectra. Therefore, we did not discuss the results in the manuscript.
- (5) We fully agree with this comment. To estimate the changes in the electronic structure of Au on the adsorption or absorption of C, the detection of a difference between the 3*s* core levels of metal centers around C and the corresponding centers in pristine gold before C insertion provides direct evidence. The given reference provides a calculation of the core levels of surface and subsurface atoms bound to C on Au(111) surfaces, and suggests that the 3*s* level of Au containing interstitial C shifts the BE to a higher value. However, to detect the 3*s* level of Au, high energy XPS with Cr K_β ($h\nu = 5946.7$ eV) is required to measure electron attenuation lengths over a kinetic energy range of ~2500 to ~5900 eV. The availability of this instrument is very limited in China (maybe fewer than 3). We contacted the owners of the instruments and were told that the 3*s* level of

a Au foil is undetectable with a commercial high energy XPS instrument. For a special measurement of the 3s level of Au, a substrate overlayer method is used in conjunction with synchrotron radiation ($h\nu = 1800\text{-}4600$ eV) excited XPS². Currently this method and this instrument is not available for our samples. Therefore, we have discussed the 3s level of Au by DFT calculations in the revised manuscript, and leave it an open question for future study.

Fig. 1. XPS spectra of (a) the Au 4f level; and (b) the C 1s level for fresh and re-used Au nanocatalysts containing interstitial carbon supported on ordered mesoporous carbon. C-Au-2.4/OMC-R represents the catalyst after reused by five runs. For comparison, commercial Au/TiO₂, Au/SiO₂ and Au/C were also investigated.

References

1. Jiang, Z., Zhang, W., Jin, L., Yang, X., Xu, F., Zhu, J., Huang, W. *J. Phys. Chem. C* **111**, 12434 (2007).
2. Diplas, S., Watts, J. F., Morton, S. A., Beamson, G., Tsakirooulos, P., Clark, D. T., Castle, J. E. *J. Electron. Spectrosc. Relat. Phenom.* **113**, 153 (2001)

We have revised our manuscript accordingly.

Revised manuscript, Page 8:

- (1) “This distinctive reversal of the binding energy shift in the Au 4f doublet as a function of Au particle size has been observed on Au/TiO₂, and assigned to a combined contribution of the charge transfer from surface to clusters and the particle size effect¹⁴.” has been changed to “This distinctive reversal of the binding energy shift in the Au 4f doublet as a function of Au particle size has been observed in Au/TiO₂ with

similarly small changes, and may be assigned to a combined contribution of the charge transfer from surface to clusters, the initial state effect and the electrostatic final state effect¹⁵.”.

Revised manuscript, Page 7, a sentence has been added:

- (2) “Fig. 2a shows measured and fitted Au 4f XPS spectra for Au nanocatalysts, and the major peaks for Au⁰ in all can be fitted with minors of Au⁺ and Au³⁺ which may originate from undercoordinated sites³⁸.”.

Revised manuscript, Page 8:

- (4) “In the present case, C diffusion that leads to the charge transfer either from the carbon atom or from the *s*, *p* electron redistribution of Au would be reasonable, and this effect is dominant for the charge gain.” has been changed to “In the present case, C diffusion that leads to a rearrangement of electron density may be dominant for the charge transfer. Viñes et al. found that the 3*s* core levels of Au around C shifts the binding energy to a higher value compared to a pristine Au(111) surface with no interstitial C, which is an evidence for the redistribution of orbitals²⁸. The high energy XPS should be used to give more distinct and direct evidence on interstitial C in future studies. Here a charge transfer either from the carbon atom or from the *s*, *p* electron redistribution of Au may be reasonable.”.

References

28. Piqué, O., Koleva, I. Z., Viñes, F., Aleksandrov, H. A., Vayssilov, G. N., Illas, F. *Angew. Chem. Int. Ed.* **58**, 1744 (2019).
38. Martínez, B., Piqué, O., Prats, H., Viñes, F., Illas, F. *Appl. Surf. Sci.* **513**, 145765 (2020).

Comment 7. Major: The shifts on the E_B can have multiple origins. Considering size effect, the smaller, the more atom-like are the NPs, and less diffuse electrons, which would make the E_B to increase. However, the charge transfer to (from) Au could decrease (increase) it. However, when having larger surface/volume ratio, one has more undercoordinated sites, and such are known to reduce the BE, as they accumulate there charge density (see doi:10.1007/s11426-010-0086-z). The discussion should be reconsidered, as, being many factors, one cannot isolate them and rule most out so to explain solely based on one.

Response 7. We agree with the comment that the shifts in binding energy can have multiple origins. In the case of Au, as mentioned by the reviewer, the negative core level shift for the surface atoms of macroscopic Au can be explained by an initial state effect due to $6s \rightarrow 5d$ charge reorganization for more undercoordinated surface atoms which accumulate their charge density¹. Such intraatomic charge transfer causes a smaller binding energy for less

coordinated Au atoms than fully coordinated bulk atoms. For the nanoclusters, the systematic energy shift can be explained by a final state effect according to the size-dependent electrostatic interaction between the ionised cluster and escaping photoelectron. The initial state shift is mainly overcompensated by the electrostatic final state effect. As a result, the binding energy increases with the surface/volume value as highlighted in comment 5, and the photoemission onset is influenced by an initial state effect involving charge transfer². The effect of the undercoordinated sites should be considered. Accordingly, we have revised our manuscript.

References

1. Zeng, Z., Ma, X., Ding, W., Li, W. *China Chem.* **53**, 402 (2010).
2. Viñes, F., Gomes, J. R. B., Illas, F. *Chem. Soc. Rev.* **43**, 4922 (2014).

Revised manuscript, Page 7, a sentence has been added:

“There are many possible reasons for the shifts in binding energy.”.

Revised manuscript, Page 7:

“This shift is attributed to the particle size effect, in good agreement with the fact that E_B shifts to a higher energy for clusters compared to that of the bulk metal and the change is a linear function of the average coordination number” has been changed to “This shift may be attributed to the particle size effect which is a final state effect according to the size-dependent electrostatic interaction between the cluster and an escaping photoelectron. It has also been reported that there is a negative core level shift for the surface atoms of macroscopic Au due to an initial state effect, which is $6s \rightarrow 5d$ charge reorganization for the more undercoordinated surface atoms; but for nanoclusters, the initial state shift is mainly overcompensated by the electrostatic final state effect⁴¹. The present positive shift is in good agreement with the fact that there are shifts to a higher energy for clusters compared to that of the bulk metal. The change is determined by the surface-to-volume ratio, and the photoemission onset is influenced by an initial state effect involving charge transfer^{40,42}.”.

References

41. Zeng, Z., Ma, X., Ding, W., Li, W. *Sci. China Chem.* **53**, 402 (2010).
42. Viñes, F., Gomes, J. R. B., Illas, F. *Chem. Soc. Rev.* **43**, 4922 (2014).

Comment 8. Minor: Authors suggest a charge transfer from C to Au. This goes along, e.g. with the charge transfer from graphene to Au, but however other reports indirectly suggest positively charged C, although with no Bader charges computed [<https://doi.org/10.1002/anie.201813037>]. That should be discussed.

Response 8. Thank you for this comment. C diffusion in the Au lattice may lead to the rearrangement of electron density that may be dominant for the charge transfer. As mentioned by the reviewer, charge transfer from C to gold or gold to C is possible since other factors for example the *s*, *p* electron redistribution of Au, are combined together. All these factors should be discussed. We have revised our manuscript accordingly.

Revised manuscript, Page 8, a reference has been added:

“In the present case, C diffusion that leads to the charge transfer either from the carbon atom or from the *s*, *p* electron redistribution of Au would be reasonable, and this effect is dominant for the charge gain.” has been changed to “**In the present case, C diffusion that leads to a rearrangement of electron density may be dominant for the charge transfer. Viñes et al. found that the 3*s* core levels of Au around C shifts the binding energy to a higher value compared to a pristine Au(111) surface with no interstitial C, which is an evidence for the redistribution of orbitals²⁸. The high energy XPS should be used to give more distinct and direct evidence on interstitial C in future studies. Here a charge transfer either from the carbon atom or from the *s*, *p* electron redistribution of Au may be reasonable.**”.

References

28. Piqué, O., Koleva, I. Z., Viñes, F., Aleksandrov, H. A., Vayssilov, G. N., Illas, F. *Angew. Chem. Int. Ed.* **58**, 1744 (2019).

Comment 9. Critical: The authors neglect the ZPE correction, but such should be easily added, as authors should vibrationally characterize the minima and transition state found. This can be also very critical in the H₂ dissociation, given its large stretching frequency.

Response 9. We are sorry for neglecting this critical point. In the revised manuscript, we have included the ZPE correction in calculations of the H₂ dissociation process. The results have been updated in Fig. 4a.

Revised manuscript, Page 14, Fig. 4a has been updated including the ZPE correction:

Fig. 4| DFT calculations. a, Energy profiles for the dissociation of H_2 at Au surfaces. The black line belongs to heterolytic dissociation on clean Au(111), and the red one to heterolytic dissociation on C-modified Au(111). **The zero-point energy correction is included in the profile.** b, Adsorption modes of 3-NS and its corresponding energies on the clean Au(111) and C-modified Au(111).

Comment 10. Major: The dynamics of the C movement to the surface or its insertion back to the Au matrix could explain the reusability? In this sense, there is a recent study on such which could address the discussion; see doi:10.1016/j.apsusc.2020.145765.

Response 10. We appreciate this comment, and indeed observed that the subsurface carbon atom on Au(111) diffuses to the surface as a result of H adsorption by DFT calculations. This phenomenon is in good agreement with the dynamic mobility of C atoms on transition metal (111) surfaces under working conditions of low C coverage¹. As a result, the presence of subsurface C would facilitate the mobility of carbon in the lattice, and possibly to some C-free areas of the surface¹. It could open another channel to move C to all possible sites in the lattice. The H adsorption would be further improved due to the increasing concentration of C on surface. In addition, the dynamics of the C movement to the surface or its insertion back to the Au matrix may also inhibit the accumulation of the organic substances, and enhance the reusability. In addition, a continuous exchange of C atoms between surface and subsurface translates into a dynamic equilibrium for Au. As a result, the dynamics of C movement to the surface or its re-insertion to the Au matrix may be responsible for a uniform expansion over the (111) plane in a such a 2 nm Au nanoparticle.

References

1. Martínez, B., Piqué, O., Prats, H., Viñes, F., Illas, F. *Appl. Surf. Sci.* **513**, 145765 (2020).

We have revised our manuscript accordingly.

Revised manuscript, Page 6:

Several sentences have been added: “It should be mentioned that the observed lattice expansion is small possibly due to the relatively low C solubility and the intrinsic lattice contraction in small size particles. However, a continuous exchange of C atoms in Au between surface and subsurface has been reported so that a dynamic equilibrium is established. The mobility of carbon in the lattice may cause residual vacancies and in turn produce a relatively uniform lattice expansion³⁸.”.

Revised manuscript, Page 13:

“It should be mentioned that the subsurface carbon atom on Au(111) diffuses to the surface as a result of H adsorption, in good agreement with the dynamic mobility of C atoms on transition metal (111) surfaces under working conditions of low C coverage³⁸. As a result, the presence of subsurface C would facilitate the mobility of carbon in the lattice, possibly to some C-free areas of the surface³⁸. This would further enhance the H adsorption.”.

Revised manuscript, Page 15:

“In addition, a continuous exchange of C atoms between surface and subsurface possibly occurs which may facilitate the adsorption of H₂³⁸. The dynamics of C movement in the Au lattice may also be responsible for the desorption of products to inhibit accumulation on the surface which enhances the reusability.”.

References

38. Martínez, B., Piqué, O., Prats, H., Viñes, F., Illas, F. *Appl. Surf. Sci.* **513**, 145765 (2020).

Comment 11. Critical: Perhaps the major critical point is the lack of description of the computational part; it is just succinctly treated, as if it was a mere and basic technique; but the details are necessary for a correct assessment; otherwise, the results could be simply wrong, either because of the model or the method. The list of open questions are, convergence of a property like the adsorption energy with respect the number of layers, basis set kinetic energy and convergence with respect it? K-points density and type, and convergence with respect it? There are more than one single type of hollows (or bridges or tops in the (211) case); have them been all explored? How have the authors treated the adsorption of 3-nitrostyrene; specifically sites and configurations. How to calculate vibrations. The optimization electronic and atomic convergence criteria. The simulation of dispersive forces? What functional have they used and why? How is the smearing treated and details. Types of

pseudopotentials. How are isolated atoms or molecules calculated? How are transition state located? With which method and details. All this could well be introduced in a section in the supplementary information.

Response 11. We appreciate this critical comment. In the revised version, we have added more detail to the Computational Details section of the Supplementary Information.

Revised Supplementary Information, Page 37:

“Computational details

All the DFT calculations were carried out by the Vienna Ab-initio Simulation Package (VASP)^{4,7}, where the GGA-PBE⁸ density functional was employed. A projected augmented wave method (PAW)^{9,10} with a cut-off energy of 400 eV was used to describe the interaction between the ionic core and valence electrons. A 5×5×1 gamma-centered k-point mesh was used for the clean surface, surface carbon and subsurface carbon modified ones, which were modeled by five-layer Au(111) and Au(211) surfaces with p(2×2) supercells, and electronic occupancies were determined by a first order Methfessel-Paxton scheme¹¹ with a smearing width of 0.2 eV. The top three layers were relaxed, while the bottom two were fixed at the bulk lattice positions. A 12 Å of vacuum layer was set between the periodically repeated slabs to avoid inter-slab interactions. The criteria for the convergence of electronic, and ionic relaxations were set to be 10⁻⁵ eV and 0.03 eV Å⁻¹, respectively. All the parameters were optimized with respect to the adsorption energy of carbon atom on the clean Au(111) surface (Supplementary Table 4). The locations of carbon atoms on the surface and subsurface were comparatively investigated for clean and C-containing Au(111) and Au(211) surfaces and subsurfaces (Supplementary Tables 5 and 6). The isolated molecules and atoms were optimized in a box of 20 Å×20 Å×20 Å with the same criterion as those for geometric optimization of the Au surfaces. The adsorption energies were calculated by $E_{\text{ads}} = E_{\text{adsorbate/surface}} - E_{\text{surface}} - E_{\text{adsorbate}}$, where $E_{\text{adsorbate/surface}}$, E_{surface} and $E_{\text{adsorbate}}$ are respectively the total energy of a surface covered with an adsorbate, the clean surface slab, and an isolated adsorbate. The transition states of hydrogen dissociation on the surfaces were located by the dimer method¹² where the criteria for the convergence of electronic, and ionic relaxations were set to be 10⁻⁶ eV and 0.025 eV Å⁻¹, respectively. The geometrical structures for the transition states were confirmed by subsequent vibrational calculations carried out using the numerical finite difference method, which generated only one imaginary frequency for a specific transition state. The zero-point energy correction was included in the calculation for the energy barrier of H₂ dissociation. The adsorption configurations and energies for 3-NS on the studied surfaces are compiled in Supplementary Tables 7 and 8.”.

Supplementary Table 4. Convergences of C adsorption on Au(111) surfaces with respect to the parameters used for DFT calculations.

K-points tests				
K-points	3×3×1	5×5×1	7×7×1	9×9×1
$E_{\text{ads, C}}$ (eV)	-4.97	-4.40	-4.55	-4.65
Cut-off energy tests				
Cut-off energy (eV)	350	400	425	450
$E_{\text{ads, C}}$ (eV)	-4.41	-4.40	-4.40	-4.40
Supercell tests				
Supercell	2×2		3×3	
$E_{\text{ads, C}}$ (eV)	-4.35		-4.40	
Vacuum layer tests				
Vacuum layer (Å)	7.0	12.0	17.0	
$E_{\text{ads, C}}$ (eV)	-4.40	-4.40	-4.40	
Ionic relaxation criterion tests				
EDIFFG (eV/Å)	0.03	0.025	0.020	
$E_{\text{ads, C}}$ (eV)	-4.40	-4.40	-4.40	
Electronic relaxation criterion tests				
EDIFF (eV)	10^{-5}	10^{-6}	10^{-7}	
$E_{\text{ads, C}}$ (eV)	-4.40	-4.40	-4.40	
Atomic layer tests				
Atomic layers	3	5	7	
$E_{\text{ads, C}}$ (eV)	-4.40	-4.40	-4.01	

Supplementary Table 5. Carbon atom at different sites of Au(111) surface and subsurface.

	Location	E_{ads} (eV)
Surface C	top site	-2.34
	fcc threefold hollow site	-4.41
	hcp threefold hollow site	-4.29
Subsurface C	tetrahedral interstice	-4.33
	octahedral interstice	-3.91

Supplementary Table 6. Carbon atom at different sites of Au(211) surface and subsurface.

	Location	E_{ads} (eV)
Surface C	Low fcc	-4.11
	Low hcp	-4.11
	High fcc	-4.42
	High hcp	-4.56
	Hollow	-4.53
	Bridge	-4.12
	Subsurface C	Low octahedral
High octahedral		-3.65
Step octahedral		-3.50
Low tetrahedral		-4.26
High tetrahedral		-4.40
Step tetrahedral		-3.94

Supplementary Table 7. Adsorption energies of 3-NS on Au(111) and C-Au(111) surfaces.

	Configuration	E_{ads} (eV)
Au(111)	O_bridge 	-0.51
	Parallel 	-0.81
	Perpendicular 	-0.41
C-Au(111)	C_top 	-3.04
	O_bridge 	-0.41
	Parallel 	-0.78
	Perpendicular 	-0.50

Supplementary Table 8. Adsorption energies of 3-NS on Au(211) and C-Au(211) surfaces.

	Configuration	E_{ads} (eV)
Au(211)	O_bridge 	-0.28
	Parallel 	-0.69
	Perpendicular 	-0.29
C-Au(211)	C_top 	-2.83
	O_bridge 	-3.07
	Parallel 	-0.51
	Perpendicular 	-0.23

References

4. Kresse, G., Hafner, J. *Phys. Rev. B* **47**, 558 (1993).
5. Kresse, G., Hafner, J. *Phys. Rev. B* **49**, 14251 (1994).
6. Kresse, G., Furthmüller, J. *Comput. Mater. Sci.* **6**, 15 (1996).
7. Kresse, G., Furthmüller, J. *Phys. Rev. B* **54**, 11169 (1996).
8. Perdew, J. P., Burke, K., Ernzerhof, M. *Phys. Rev. Lett.* **77**, 3865 (1996).
9. Blöchl, P. E. *Phys. Rev. B* **50**, 17953 (1994).
10. Kresse, G., Joubert, D. *Phys. Rev. B* **59**, 1758 (1999).
11. Methfessel, M., Paxton, A. T. *Phys. Rev. B* **40**, 3616 (1989).
12. Henkelman, G., Jónsson, H. *J. Chem. Phys.* **111**, 7010 (1999).

Comment 12. Minor: Put commas between all items in a sequence. In page 12, the sentence with “activated by the only the C atom, and both” reads strange; please amend. In the computational details, it should read p(2×2), lack of dimensionality (better explicit vacuum in Angstroms).

Response 12. Thank you for this comment.

- (1) Commas have been added between all items in a sequence in the revised manuscript.
- (2) We are sorry for the typo. It should be “the H₂ can be activated only by the C atom, and both...”
- (3) We appreciate the suggestion. In the computational details, we have corrected the inappropriate expression of p(2*2) to p(2×2). The disruption for the vacuum layer has also been corrected to “a vacuum layer of 12 Å”.

Revised manuscript, Page 13:

“When carbon diffuses into the Au(111) surface, the H₂ can be activated by the only the C atom, and both H atoms are adsorbed on the C atom, reducing the activation barrier to 0.60 eV, implying a lower energy process.” has been changed to “When carbon diffuses into the Au(111) surface, the H₂ can be activated only by the C atom, and both H atoms are adsorbed on the C atom, reducing the activation barrier to 0.59 eV, implying a lower energy process.”.

Revised manuscript, Page 18:

“The Au(111) and (211) surfaces were modelled using a seven-layer metal slab describing a p(2*2) unit cell and seven vacuum layers. The relative positions of the atoms were as in the bulk, with an optimized lattice

parameter of 3.8543 Å (exptl 3.8034)⁶¹. Three different surface sites (top, bridge, and hollow) and two different subsurface sites (tetrahedral and octahedral) were considered. Carbon atoms were placed on the surface and/or in subsurface sites on both sides of the slab. All atoms in the unit cell were relaxed in all directions. DFT calculations were carried out with the Vienna ab initio simulation package (VASP)^{62,63}, which performs an iterative solution of the Kohn-Sham equations in a plane-wave basis set.” has been changed to “DFT calculations were carried out with the Vienna ab initio simulation package (VASP)^{66,67}. The Au(111) and (211) surfaces were modelled using a five-layer metal slab describing a p(2×2) unit cell and a vacuum layer of 12 Å. The relative positions of the atoms are as in the bulk, with an optimized lattice parameter of 4.17 Å, which is close to the previously experimental and the calculated ones (i.e., 4.08 and 4.17 Å, respectively)⁶⁸. Three different surface sites (top, bridge, and hollow) and two different subsurface sites (tetrahedral and octahedral) were considered. More details are presented in the supplementary information (Computational details and Tables 4-8).”.

Reviewer #2

Comment. This is a potentially interesting paper but the catalysis needs a lot of attention. The claim is that by making the catalyst this way they make a material that has interstitial C in Au nanoparticles supported on carbon. They then carry out a standard model hydrogenation reaction (so there is no novelty in the catalysed reaction). I am not commenting on the microscopy or the DFT calculations.

Response. We thank you for reviewing our work on a new catalytic material that has interstitial C in Au nanoparticles supported on carbon. The novelty is the formation of interstitial C in the Au lattice, and the charge transfer between the C and Au which has not been reported to the best of our knowledge. A standard model hydrogenation reaction was chosen because a very weak catalytic hydrogenation activity was observed over gold loaded on a less-active support such as carbon and silica in the literature. Therefore, it is acceptable that the high activity and selectivity for the present interstitial catalyst originates from the charge transfer between interstitial C and Au. We are now using our novel catalysts in some challenging reactions, and shall report the results in the future.

Comment 1. The catalysis they observe with their catalyst is certainly interesting. But the main claim is that their catalyst is far better than gold supported on carbon, silica or titania. But the comparison is with three commercial catalysts and we have no idea how they are made and what they comprise. For a paper at this level this is unacceptable. They need to prepare their own Au comparison catalysts so that the particle sizes are similar and then do a comparison.

Response 1. We thank you for the comment that the catalysis part is certainly interesting. The main doubt is the referenced commercial catalysts. The commercial catalysts are products from Haruta Gold Inc. (Japan). The company was founded by Haruta Gold Incorporated, and most product details have been published, for example, Au/C in *Appl. Catal. A Gen.* **369**, 8 (2009) and *Appl. Catal. A Gen.* **377**, 42 (2010); Au/TiO₂ in *J. Catal.* **144**, 175 (1993). The reason for using commercial products is that they can be easily purchased by researchers for comparison. By comparing the XPS spectra, XRD patterns and TEM images for their customized samples with the literature, we believed that these reference samples satisfy our requirements, i.e. relatively uniform nanoparticle sizes around 4 nm. Please see their provided data below (Tables 1 - 3 and Figs. 1 - 6).

Table 1. Physico-chemical Properties of Au/TiO₂

Physico-chemical Properties	Results	Methods
Diameter of support	9.4 μm (mean diameter) 5.5 μm (median size) 5.5 μm (modal diameter)	Laser Diffraction
Specific surface area	54 m ² /g	BET
Content of gold	0.96 wt%	ICP
TEM observation	5 figures	
Diameter of gold	mean diameter 4.1 nm standard deviation 2.2 nm	HAADF-STEM
CO oxidation in gas phase T _{1/2} , Temp. for 50% conversion (water at T _{1/2} , ppm)	a) -6 °C (H ₂ O: 11 ppm) c) 3 °C (H ₂ O: 50 ppm) d) 4 °C (H ₂ O: 50 ppm) e) 7 °C (H ₂ O: 63 ppm) f) 10 °C (H ₂ O: 126 ppm)	Fixed Bed Reactor

Fig. 1. (a) TEM, (b-e) HAADF STEM images and (f) distribution of the Au particle size diameter for a commercial Au/TiO₂ catalyst.

CO oxidation in gas phase

<Pretreatment> air(N₂:O₂ = 4:1) : 50 mL/min, 250 °C, 1h.

<Catalytic tests> sample amount : 150 mg, 1% CO in air : 50 mL/min., SV = 20,000 mL/h · g_{cat}.

CO concentration was measured by gas chromatography.

Fig. 2. Temperature dependence of CO conversion in CO oxidation for Au/TiO₂.

Table 2. Physico-chemical Properties of Au/SiO₂

Physico-chemical Properties	Results	Methods
Diameter of support	51.6 μm (mean diameter) 52.1 μm (median size) 55.0 μm (modal diameter)	Laser Diffraction
Specific surface area	294 m ² /g	BET
Content of gold	1.07 wt%	AAS
TEM observation	5 figures	
Diameter of gold	mean diameter 5.8 nm standard deviation 2.7 nm	HAADF-STEM
CO oxidation in gas phase Conversion at 250 °C (water at 250 °C, ppm)	a) 9.2% (H ₂ O: 56 ppm)	Fixed Bed Reactor

Fig. 3. (a) TEM, (b-e) HAADF STEM images and (f) distribution of the Au particle diameter for a commercial Au/SiO₂ catalyst.

CO oxidation in gas phase

<Pretreatment> air(N₂:O₂ = 4:1) : 50 mL/min, 250 °C, 30 min.

<Catalytic tests> sample amount : 150 mg, 1% CO in air : 50 mL/min., SV = 20,000 mL/h · g_{cat}.
CO concentration was measured by gas chromatography.

Fig. 4. Temperature dependence of CO conversion in CO oxidation for Au/SiO₂.

Table 3. Physico-chemical Properties of Au/C.

Physico-chemical Properties	Results	Methods
Diameter of support	7.1 μm (mean diameter) 6.7 μm (median size) 7.1 μm (modal diameter) (in ethanol)	Laser Diffraction
Specific surface area	731 m ² /g	BET
Content of gold	0.89 wt%	ICP
TEM observation	5 figures	
Diameter of gold	mean diameter 6.3 nm standard deviation 1.6 nm	HAADF-STEM
CO oxidation in gas phase Conversion at 250 °C (water at 250 °C, ppm)	a) 0 °C (H ₂ O: 8 ppm) b) 0 °C (H ₂ O: 23 ppm) c) 0 °C (H ₂ O: 63 ppm)	Fixed Bed Reactor

Fig. 5. (a,b) TEM, (c-e) HAADF STEM images and (f) distribution of the Au particle diameter for the commercial Au/C catalyst.

CO oxidation in gas phase

<Pretreatment> air($N_2:O_2 = 4:1$) : 50 mL/min, 250 °C, 30 min.

<Catalytic tests> sample amount : 150 mg, 1% CO in air : 50 mL/min., SV = 20,000 mL/h · g_{cat.}.

CO concentration was measured by gas chromatography.

Fig. 6. Temperature dependence of CO conversion in CO oxidation for Au/C.

Actually, we also synthesized Au/TiO₂, Au/SiO₂ and Au/C reference samples with our mesoporous carbon carriers using the deposition-precipitation or deposition-reduction method. TEM images of the synthesized catalysts are shown in Fig. 7. The catalytic performance in the selective hydrogenation of 3-vinylaniline for these catalysts was similar to the customized ones supported on the same carrier, and also compared to the well-documented literature. For example, the TOF value for our mesoporous Au/TiO₂ was estimated to be 226 h⁻¹, and both Au/C and Au/SiO₂ were almost inactive to this reaction. However, the synthesis of these samples was complicated. It has been reported that the nanoparticle sizes and crystal phases of transitional oxide carriers will for certain affect the catalytic performance of gold nanoparticles. Since commercial products are easily obtained for comparison, we decided to purchase customized samples for our research.

Fig. 7. TEM images for (a) Au/TiO₂, (b) Au/SiO₂ and (c) Au/C.

Comment 2. The loss of gold can be a real problem and they need to show the data for Au retained on the catalyst after use. Typically quite a lot can be solubilised but it plates out on the reactor and so is not observed in solution.

Response 2. We fully agree with the reviewer that the loss of gold can be a real problem for a liquid reaction. Several tests have been performed to confirm the negligible leaching of gold for the present catalyst.

- (1) A solid quenching test: A solid quenching test was performed using mercapto functional group-containing mesoporous silica SH-SBA-15 as the trapping agent. Once the metal species leach into solutions, mercapto functional group will trap them, and quench the catalysis by leaching metal species in liquid reactions. The unchanged hydrogenation activity and selectivity in the presence of SH-SBA-15 demonstrates there is no leaching of gold with the present C-Au/OMC catalysts regardless of Au particle size, and the solubilized gold in solution during reaction and redeposition on the carrier surface after reaction can be excluded (Fig. 1).
- (2) Reusability: Both the initial reaction rate (r_0) and turn over number (TON) in ten successive runs remained almost the same (Fig. 2), indicating the number of active centers does not significantly change.
- (3) XPS spectra: The XPS spectrum was taken for the re-used catalyst, and shows indistinct changes compared with the fresh catalyst (Fig. 3).
- (4) TEM image: the average Au particle size for the re-used catalyst also remained almost unchanged, as shown by TEM images (Fig. 4).
- (5) Au concentration measurement: The Au concentration in the reused catalyst after 10 runs remained at 0.91 wt%, similar to that determined for the fresh catalyst.

In these cases, the leaching of gold nanoparticles can be excluded. A surface reaction on the Au nanoparticles instead of a reaction in solution catalyzed by leached Au is the best model for the present reaction.

Fig. 1. Trapping test. Comparison of the conversion of 3-NS in the absence and presence of the solid trapping agent SH-SBA-15 over C-Au nanocatalysts. Reaction conditions: 0.78 - 1.87 μmol of Au, 43.68 - 104.72 mg SH-SBA-15, 0.41 mmol of 3-NS, 5 mL of ethanol, 140 °C, 4.0 MPa H₂.

Fig. 2. Reusability in terms of the initial reaction rate (r_0) and turn over number (TON) in successive runs with the recovered C-Au-2.4/OMC catalyst in the selective hydrogenation of 3-NS. Reaction conditions: 20 mg of C-Au-2.4/OMC, 0.41 mmol of substrate, 5 mL of ethanol, 140 °C, 800 rpm, 4.0 MPa H₂.

Fig. 3. XPS spectra of the Au 4f level for fresh and re-used Au nanocatalysts containing interstitial carbon supported on ordered mesoporous carbon. C-Au-2.4/OMC-R represents the catalyst after reused by five runs. For comparison, commercial Au/TiO₂, Au/SiO₂ and Au/C were also investigated.

Fig. 4. TEM images for the (a) fresh C-Au-2.4/OMC catalyst and (b) used C-Au-2.4/OMC-R10 catalyst. The insets are particle size histograms of the Au nanoparticles measured from at least 200 nanoparticles.

Revised manuscript, Page 11:

The paragraph has been re-written:

“The leaching of gold species from the solid catalyst which can catalyze the hydrogenation has been argued by some researchers. A solid quenching test was performed using mesoporous silica SH-SBA-15 containing mercapto functional groups as the trapping agent. Once the metal species leach into solution, mercapto functional group will trap them, and quench the catalysis by leaching metal species in liquid reactions⁵⁸. The unchanged hydrogenation activity and selectivity in the presence of SH-SBA-15 demonstrates there is no leaching of gold with the present C-Au/OMC catalysts regardless of Au particle size, and solubilized gold in solution during the reaction and redeposition on the carrier surface after the reaction can be excluded (Supplementary Fig. 16). The reusability of C-Au-2.4/OMC was also tested. Both the initial reaction rate (r_0) and turn over number (TON) in ten successive runs remained almost the same (Supplementary Fig. 17), indicating the number of active centers does not significantly change. The re-used catalysts were characterized. The XPS spectrum shows no distinct changes compared with the fresh catalyst. The Au concentration in the reused catalyst after 10 runs remained at 0.91 wt%, also similar to that determined for the fresh catalyst. In addition, the average Au particle size for the re-used catalyst also remained almost unchanged, as shown by TEM images (Supplementary Fig. 18).”.

Comment 3. Finally the English needs attention.

Response 3. Thanks for this comment. The manuscript has been revised by a professor who is a native English speaker, and some typos have been corrected.

Reviewer #3

Comment. The authors claim to have supposedly made a catalyst material comprised of gold particles embedded in mesoporous carbon with carbon atoms occupying interstitial sites in the gold lattice. Furthermore, they claim this interstitial doped catalyst is superior to Au/TiO₂ for the chemoselective hydrogenation of 3-nitrostyrene. Several things concern me about the microstructural characterization aspects of this paper.

Response. We thank you for the comment.

Comment 1. Firstly, the Au Particle size distributions are not ‘monodisperse’ as repeatedly claimed in the text as proven by their own particle size distributions.

Response 1. We thank for this comment, and have removed the description of “monodisperse” in the whole manuscript.

Revised manuscript, Page 4:

“Here, monodispersed gold nanoparticles with carbon atoms occupying interstitial sites in the lattice (C-Au)” has been changed to “Here, uniform gold nanoparticles with carbon atoms occupying interstitial sites in the lattice (C-Au)”.

Revised manuscript, Page 5:

“Monodispersed nanoparticles with uniform sizes can be clearly observed in large domains in the high-angle annular dark-field scanning transmission electron microscope (HAADF-STEM) images (Supplementary Fig. 4).” has been changed to “Nanoparticles with uniform sizes can also be clearly observed in large areas in high-angle annular dark-field spherical aberration corrected-scanning transmission electron microscope (HAADF-ACSTEM) images (Figs. 1c,d and Supplementary Fig. 4).”.

Comment 2. Secondly, it has not been proven that the Au particles are actually within the pores of the activated carbon spheres. TEM/STEM is a 2D projection technique unless applied in tomography mode. Indeed, several of the supplementary figures show gold particles in profile on the C surface. Furthermore, if the gold particles were really embedded in the interior of the activated C spheres how do the generated photoelectrons escape in the XPS experiments?

Response 2. Thanks for this comment. First, we are sorry for the misunderstanding. The gold nanoparticles are not fully embedded in the pore walls of the mesoporous carbon spheres. As mentioned by the reviewer, if the gold particles are fully embedded in the pore walls, the generated photoelectron signal will be extremely weak in the XPS experiments, as evidenced by our previous report on NiO which are fully embedded in mesoporous carbon¹. In addition, we found that if the nanoparticles are partially exposed to the pore space, the generated photoelectrons can escape in the XPS experiments but the detected concentration is much lower than the ideal concentration². Similar results have been found for the present catalysts. Therefore, the nanoparticles are partially embedded into the mesopore walls with partial exposure to the pore space.

Second, we fully agree that TEM/STEM is a 2D projection technique. We conclude that the gold nanoparticles are located inside the mesoporous carbon spheres instead of being stuck to the outer shell for the following reasons: (1) The supplementary TEM and AC-TEM images containing nanoparticles in the original submission show a change in contrast, implying they are located at different depths. (2) If we make a rough calculation of the weight

of a carbon sphere with a diameter of 100 nm, there will be approximately 240 gold nanoparticles with a 2 nm in diameter for a 1:100 weight ratio. If all nanoparticles are uniformly distributed on the outside of the sphere, the projected image would show a high concentration of NPs at the edges of the projected circle and isolated nanoparticles in the centers of the circles. If the nanoparticles are uniformly distributed throughout the mesoporous carbon sphere, a higher concentration in the center of the circle would be seen. Therefore, we took the HAADF-STEM image of the C-Au-2.4/OMC catalyst. There are more bright particles in the centers of the circles than at the edges. (3) We have earlier shown by TEM and SEM that the nanoparticles are inside the mesostructure instead of being stuck to the surface of the carbon spheres. For example, images of an ultrathin section of AuPd nanoalloy supported on ordered mesoporous carbon (AuPd/OMC) are quite similar to those of the bulk catalyst, showing that the metal nanoparticles are partially embedded in the mesopore walls instead of coating the outer surface of the carbon support (Fig. 2)². Interestingly, partially exposed nanoparticles can be clearly observed. Similar results have also been found in the present Au nanoparticles in ordered mesoporous carbon (Fig. 3).

Last but not least, the outer surface area of the sphere is as low as 23 m²/g. On consideration of the coordination-assisted self-assembly approach and the high surface areas of mesoporous structure of the carbon sphere (594 m²/g), we conclude that the gold nanoparticles are located inside the structure during the assembly and carbonization. Gold nanoparticles are partially embedded in the carbon pore walls, and partially exposed to the pore space.

Fig. 1. (a) TEM and (b) HAADF-STEM images for C-Au-2.4/OMC.

Fig. 2. Representative TEM image of the (a) bulk and (b) ultrathin sections of AuPd/OMC. Inset b is a high magnification image.

Fig. 3. TEM images and particle size distribution obtained from at least 200 nanoparticles for C-Au-3.9/OMC.

References

1. Wang, W., Wang, H., Wei, W., Xiao, Z., Wan, Y. *Chem. Eur. J.* **17**, 1346 (2011).
2. Zhu, X., Guo, Q., Sun, Y., Chen, S., Wang, J., Wu, M., Fu, W., Tang, Y., Duan, X., Chen, D., Wan, Y. *Nat. Commun.* **10**, 1428 (2019).

Accordingly, we have revised our manuscript.

Revised manuscript, Page 2:

“Here we use gold catalyst particles embedded in mesoporous carbon with carbon atoms occupying interstitial sites in the gold lattice.” has been changed to “Here we use gold catalyst particles partially embedded in the pore walls of mesoporous carbon with carbon atoms occupying interstitial sites in the gold lattice.”.

Revised manuscript, Page 5:

“Monodispersed nanoparticles with uniform sizes can be clearly observed in large domains in the high-angle annular dark-field scanning transmission electron microscope (HAADF-STEM) images (Supplementary Fig. 4).” has been changed to “Nanoparticles with uniform sizes can also be clearly observed in large areas in high-angle annular dark-field spherical aberration corrected-scanning transmission electron microscope (HAADF-ACSTEM) images (Figs. 1c,d and Supplementary Fig. 4). HAADF-STEM images were also taken for C-Au-2.4/OMC (Supplementary Fig. 5).”.

Revised manuscript, Page 5, two sentences have been added:

“Interestingly, semi-exposed gold nanoparticles can be clearly observed. Similar results have also been found in AuPd nanoparticles partially embedded in the pore walls of ordered mesoporous carbon³¹.”

“There are more bright particles in the centers of the circles than at the edges, implying the presence of nanoparticles inside the spheres.”

Revised Supplementary Information, Page 8:

Supplementary Fig. 5. (a) TEM and (b) HAADF-STEM images for C-Au-2.4/OMC.

Comment 3. Finally, and even more concerning to me, is the lack of any convincing evidence that C exists as an interstitial species in the *f.c.c.* Au lattice. Measurement of Au (111) spacings by STEM HAADF imaging over six or so lattice planes to a 0.01 Å accuracy is simply not a realistic or feasible proposition. The authors should carefully evaluate the intrinsic errors associated with making such measurements.

Response 3. We appreciate this comment. As mentioned in the literature¹, the ex situ surface and subsurface C detection on Au regular surfaces is a challenging task. The in situ detection, for example ambient pressure X-ray photoemission spectroscopy (APXPS) may be useful for the detection of such C species in noble-metal systems. However, for the nanoparticles, we have made several attempts, and the lattice expansion and electronic structure are the most repeatable attempts. The lattice expansion is reliable to some extent because:

(1) It is well-known that lattice contraction occurs in small nanoparticles or clusters, and is related to a higher proportion of outer shell undercoordinated atoms with respect to fully coordinated inner shell atoms which is proportional to the reciprocal of the particle size (Table 1). For example, about 3.0% contraction is found for an Au nanoparticle with 1.6 nm in size (in comparison with the bulk Au with a $d(111)$ of 2.350 Å)². Therefore, the observed small lattice expansion may be evidence for the interstitial C. The smaller expansion of the C-Au catalyst compared to bulk Au is possibly due to the overall relatively low C solubility and the intrinsic lattice contraction in small nanoparticles.

Table 1. Lattice spacing in gold nanoparticles². For comparison, the measured $d(111)$ value for Au nanoparticles containing interstitial C are also provided.

Particle size of Au (nm)	1.6	1.9	2.4
$d(111)$ of Au (Å) calculated ²	2.278	2.284	2.300
$d(111)$ of C-Au (Å) measured in this work	2.390	2.380	2.360

(2) A continuous exchange of C atoms in Au between surface and subsurface has been reported so that a dynamic equilibrium is established. The mobility of carbon in the lattice may cause residual vacancies and in turn a relatively uniform lattice expansion³. The final state for the dynamic C moieties would be to aggregate in graphite or amorphous carbon phases, similar to our experimental observations¹.

We must keep in mind that direct evidence that C exists as an interstitial species in the *f.c.c.* Au lattice is very important and deserves further thorough investigation. For example, the detection of a difference between the

3s core levels of metal centers around C and the corresponding centers in pristine gold before C insertion gives direct evidence.

References

1. Piqué, O., Koleva, I. Z., Viñes, F., Aleksandrov, H. A., Vayssilov, G. N., Illas, F. *Angew. Chem. Int. Ed.* **58**, 1744 (2019).
2. Wang, M., Qi, W. *J. Nanopart. Res.* **7**, 51 (2005).
3. Martínez, B., Piqué, O., Prats, H., Viñes, F., Illas, F. *Appl. Surf. Sci.* **513**, 145765 (2020).

Accordingly, we have revised our manuscript.

Revised manuscript, Page 6:

Several sentences have been added: “It should be mentioned that the observed lattice expansion is small possibly due to the relatively low C solubility and the intrinsic lattice contraction in small size particles. However, a continuous exchange of C atoms in Au between surface and subsurface has been reported so that a dynamic equilibrium is established. The mobility of carbon in the lattice may cause residual vacancies and in turn produce a relatively uniform lattice expansion³⁸.”.

Revised manuscript, Page 8:

Several sentences have been added: “In the present case, C diffusion that leads to a rearrangement of electron density may be dominant for the charge transfer. Viñes et al. found that the 3s core levels of Au around C shifts the binding energy to a higher value compared to a pristine Au(111) surface with no interstitial C, which is an evidence for the redistribution of orbitals²⁸. The high energy XPS should be used to give more distinct and direct evidence on interstitial C in future studies. Here a charge transfer either from the carbon atom or from the *s*, *p* electron redistribution of Au may be reasonable.”.

Comment 4. Furthermore, even a (very high) 2 at% C doping level in a 2 nm Au particle would still only mean the incorporation 1-2 interstitial C atoms at most. How would this cause such a uniform expansion over (111) planes in a such 2 nm Au nanoparticle?

Response 4. Thanks so much for your valuable comment.

(1) It should be mentioned that 2 at% C doping was cited from the reference [1].

The solubility (S) of C in a particle with radius r is estimated to be

$$S = S_0 \exp(2\sigma V/kTr)$$

where S_0 is the C solubility in bulk Au, defined as the ratio of the amounts of solute and solvent, σ is the surface tension, V is the volume of a metal atom, k is the Boltzmann constant, and T is the melting temperature in bulk phase. Here, $S_0 = 0.08$ at%, $\sigma = 1.42$ J m⁻², $V = 1.69 \times 10^{-29}$ m³, $k = 1.38 \times 10^{-23}$ J K⁻¹, $T = 1337.58$ K and $r = 0.8, 0.95, 1.2, 1.95$ and 4.5 nm. For σ and S_0 , the values in refs [2] and [3] were used.

The calculation used here is to show the significant increase of C solubility in the Au lattice with a decrease in particle diameter. The ideal solubility may not exactly follow the calculation. For example, if we use this formula to calculate the C solubility in Pd, it will be about 1.77 at% for a particle with a diameter of 2.5 nm. Here, $S_0 = 0.6$ at%, $\sigma = 1.482$ J m⁻², $V = 1.15 \times 10^{-29}$ m³, $k = 1.38 \times 10^{-23}$ J K⁻¹, $T = 1825.15$ K and $r = 1.25$ nm. The S_0 value was taken from reference [4]. However, several reports have found about 10 at% C can be inserted into the Pd lattice⁵. Therefore, the C doping concentration in Au may not be as low as indicated by the calculations. To avoid this misunderstanding, we have removed the detailed data on C solubility.

- (2) We fully understand your doubts about the insertion of C in the Au lattice. As mentioned by Reviewer #1, there is a rapid transition between the surface and subsurface C atoms⁶. A continuous exchange of C atoms between the surface and subsurface translates into a dynamic equilibrium for Au. The presence of subsurface C would facilitate the mobility of carbon in the lattice and possibly to some C-free areas of the surface. The final state for the dynamic C moieties would be to aggregate in graphite or amorphous carbon phases, similar to our experimental observations. Therefore, the dynamics of the C movement to the surface or its re-insertion in the Au matrix may be responsible for a uniform expansion over (111) planes in a such 2 nm Au nanoparticle.

References

1. Takagi, D., Kobayashi, Y., Hibino, H., Suzuki, S., Homma, Y. *Nano Lett.* **8**, 3 (2008).
2. Sambles, J. R. *Proc. R. Soc. London, Ser A* **324**, 339 (1971).
3. Okamoto, H., Massalski, T. B. *Bull. Alloy Phase Diagrams* **5**, 378 (1984).
4. Siller, R. H., Oates, W. A., Mclellan, R. B. *J. Less-Common Metals* **16**, 71 (1968).
5. Beck, M., Ellner, M., Mittemeijer, E. J. *Acta mater.* **49**, 985 (2001).
6. Martínez, B., Piqué, O., Prats, H., Viñes, F. & Illas, F. *Appl. Surf. Sci.* **513**, 145765 (2020).

Accordingly, we have supplied the discussion in the revised manuscript.

Several sentences have been added:

Revised manuscript, Page 6:

“It should be mentioned that the observed lattice expansion is small possibly due to the relatively low C solubility and the intrinsic lattice contraction in small size particles. However, a continuous exchange of C atoms in Au between surface and subsurface has been reported so that a dynamic equilibrium is established. The mobility of carbon in the lattice may cause residual vacancies and in turn produce a relatively uniform lattice expansion³⁸.”.

Revised manuscript, Page 13:

“It should be mentioned that the subsurface carbon atom on Au(111) diffuses to the surface as a result of H adsorption, in good agreement with the dynamic mobility of C atoms on transition metal (111) surfaces under working conditions of low C coverage³⁸. As a result, the presence of subsurface C would facilitate the mobility of carbon in the lattice, possibly to some C-free areas of the surface³⁸. This would further enhance the H adsorption.”.

Revised manuscript, Page 15:

“In addition, a continuous exchange of C atoms between surface and subsurface possibly occurs which may facilitate the adsorption of H₂³⁸. The dynamics of C movement in the Au lattice may also be responsible for the desorption of products to inhibit accumulation on the surface which enhances the reusability.”.

References

38. Martínez, B., Piqué, O., Prats, H., Viñes, F., Illas, F. *Appl. Surf. Sci.* **513**, 145765 (2020).

REVIEWERS' COMMENTS:

Reviewer #1 (Remarks to the Author):

The authors have fully addressed all my questions; furthermore, I find the explanations to other reviewers convincing (like, why to use commercial samples, helping reproducibility). I am glad to recommend publication as is.

Reviewer #2 (Remarks to the Author):

The authors have answered my comments satisfactorily

Reviewer #3 (Remarks to the Author):

The authors have made a substantial and concerted effort to answer the various comments and concerns raised by the three referees. They have also suitably modified the text and added more back-up materials to the ESI document to support their arguments. In my opinion the paper is now suitable for publication as a Nature Communication.